# Structural basis of antiphage immunity generated by a prokaryotic Argonaute-associated SPARSA system

Xiangkai Zhen [1,4], Xiaolong Xu [2,4], Le Ye [1,4], Song Xie [3,4], Zhijie Huang [1,4], Sheng Yang[1], Yanhui Wang[2], Jinyu Li [3] ✉, Feng Long [2] ✉ & Songying Ouyang [1] ✉

Argonaute (Ago) proteins are ubiquitous across all kingdoms of life. Eukaryotic Agos (eAgos) use small RNAs to recognize transcripts for RNA silencing in eukaryotes. In contrast, the functions of prokaryotic counterparts (pAgo) are less well known. Recently, short pAgos in conjunction with the associated TIR or Sir2 (SPARTA or SPARSA) were found to serve as antiviral systems to combat phage infections. Herein, we present the cryo-EM structures of nicotinamide adenine dinucleotide (NAD$^+$)-bound SPARSA with and without nucleic acids at resolutions of 3.1 Å and 3.6 Å, respectively. Our results reveal that the APAZ (Analogue of PAZ) domain and the short pAgo form a featured architecture similar to the long pAgo to accommodate nucleic acids. We further identified the key residues for NAD$^+$ binding and elucidated the structural basis for guide RNA and target DNA recognition. Using structural comparisons, molecular dynamics simulations, and biochemical experiments, we proposed a putative mechanism for NAD$^+$ hydrolysis in which an H186 loop mediates nucleophilic attack by catalytic water molecules. Overall, our study provides mechanistic insight into the antiphage role of the SPARSA system.

Argonaute (Ago) proteins are widely distributed across all kingdoms of life, ranging from bacteria to eukaryotes. Eukaryotic Ago (eAgo) proteins are well-documented as essential components of the RNA-induced silencing complex (RISC). eAgo can use small noncoding RNA molecules as a guide to direct the RISC complex towards complementary RNA targets, facilitating the cleavage of target RNA[1]. A highly conserved feature of Ago homologs is the bilobed architecture, which consists of four signature domains, including the N-terminal, PAZ (Piwi-Argonaute-Zwille), MID (Middle), and PIWI (P-element induced wimpy testis), which are connected through two structured linkers (L1 and L2)[2]. A central tunnel of eAgo is formed by N-PAZ and MID-PIWI lobes connected by L2 linkers, which cradles the guide RNA (gRNA) and the complementary target RNA[3]. The N domain acts as a wedge and helps the duplex unwind[4]. The PAZ domain facilitates the binding of the 3'-end of the gRNA via a hydrophobic pocket[5,6]. The Rossmann-like fold of the MID domain can recognize the 5'-end of the guide strands via interactions with the 5'-phosphate (5'-P)[7]. The PIWI domain consists of an RNase H catalytic fold with the DEDX tetrad (in which X is commonly His or Asp), which is responsible for target cleavage[3].

The homologs of Ago family proteins are also extremely abundant in prokaryotes (pAgo), and studies on TtAgo from *Thermus*

[1]Key Laboratory of Microbial Pathogenesis and Interventions of Fujian Province University, the Key Laboratory of Innate Immune Biology of Fujian Province, Biomedical Research Center of South China, Key Laboratory of OptoElectronic Science and Technology for Medicine of the Ministry of Education, College of Life Sciences, Fujian Normal University, Fuzhou 350117, China. [2]Key Laboratory of Combinatorial Biosynthesis and Drug Discovery (Ministry of Education), School of Pharmaceutical Sciences, Department of Neurosurgery, Zhongnan Hospital of Wuhan University, Wuhan, China Wuhan University, Wuhan 430071, China. [3]College of Chemistry, Fuzhou University, 350116 Fuzhou, China. [4]These authors contributed equally: Xiangkai Zhen, Xiaolong Xu, Le Ye, Song Xie, Zhijie Huang. ✉e-mail: j.li@fzu.edu.cn; longfe@whu.edu.cn; ouyangsy@fjnu.edu.cn

*thermophilus*[8], RsAgo from *Rhodobacter sphaeroides* and CbAgo from *Clostridium butyricum*[9–11] suggested that certain pAgos are involved in host defense by DNA-guided DNA interference in contrast to eAgo[11–13]. However, the biological functions of most pAgo proteins remain largely undiscovered in comparison with their eukaryotic counterparts[14]. Compared to eAgos, pAgos show more diversity in sequence and domain compositions, which can be subdivided into long-A, long-B and short pAgos based on their phylogeny[14–16]. The prokaryotic long-A and long-B Agos are akin to eAgos, which possess similar structural segments, including the N, PAZ, MID and PIWI domains[17,18]. In contrast, short pAgos retain the MID and PIWI domains only[19–22]. Additionally, almost all short pAgos (~60%) are inactive as the PIWI responsible for target degradation lacks a catalytic tetrad[14].

The genes that colocalize with short pAgo frequently encode proteins carrying a TIR or Sir2 domain fused to an APAZ domain in the C-terminus[14,16,19]. Recent findings have revealed that the conserved Sir2 and TIR domains possess NADase activities[23–25]. The exact biological functions of the operon in short pAgo systems have been experimentally revealed, and these operons are involved in phage resistance by inhibiting bacterial growth[20–22,26]. Although the short pAgo systems are highly diverse, they can be grouped into four phylogenetic subclades based on the features of their associated proteins[20]. Various strategies are employed by the short pAgo systems to defend against phage infections; for instance, in the SPARTA (TIR-APAZ-pAgo) and SPARSA (Sir2-APAZ-pAgo) systems, the essential cellular NAD⁺ molecules are exhausted by the accompanying Sir2 and TIR effectors[20,21], whereas the integrity of the cell membrane is disrupted by the three-gene short pAgo system (SiAgo) in *Sulfolobus islandicus*[22].

Although roles in antiviral defense cooperatively completed by short pAgo and their accessory cognate effector enzymes have been established[21,22], the detailed molecular mechanisms underlying their assembly and the enzymatic activation regulated by the recognition of invading viruses should be explored. In this study, we investigate the newly characterized SPARSA phage defense system from *Geobacter sulfurreducens* (GsSir2/pAgo) and describe the cryo-electron microscopy (cryo-EM) structures of NAD⁺-bound complexes of SPARSA in the absence or presence of gRNA and tDNA. Our results show the dedicated interactions between GsSir2 and short pAgo, revealing the structural basis of NAD⁺ recognition and the molecular mechanism underlying the recognition of gRNA target binding, which involves short pAgo and the APAZ domain of Sir2-APPAZ. Together with molecular dynamic simulations, the results indicated that the subtle structural movement of the H186-containing loop in GsSir2 is essential for NADase activation by facilitating the opening of a water channel. Taken together, our results provide the detailed molecular mechanism that underlies the antiphage defense of the SPARSA system.

## Results

### Architecture of the SPARSA phage defense system

The SPARSA antiphage defense system is encoded by an operon that consists of a short pAgo and an associated effector with an N-terminal Sir2 NADase fused to a C-terminal APAZ domain, which can form a stable heterodimeric complex in vivo and in vitro (Fig. 1a)[21]. To determine the structural basis for the interactions between GsSir2 and pAgo in the SPARSA system, we used single-particle cryo-EM analysis to determine the structure of the SPARSA heterodimer. The pure soluble SPARSA binary complex was obtained by coexpression of GsSir2 and pAgo in *E. coli*[21] and subsequent purification using Ni affinity chromatography and size exclusion chromatography (SEC) (Supplementary Fig. 1a, b).

Finally, we obtained a three-dimensional reconstruction of SPARSA at an overall resolution of 3.6 Å (Supplementary Fig. 2a–c, Table 1). The overall structure of SPARSA contains GsSir2 and pAgo with a 1:1 stoichiometry (Fig. 1b), in which pAgo is closely enveloped by the individual Sir2 and APAZ domains, appearing as an armchair shape with dimensions of ~110 Å in length and ~70 Å in width (Fig. 1b, c). The N-terminal Sir2 domain and the C-terminal APAZ domain collectively form arms that wrap the short pAgo in the middle, which resembles the

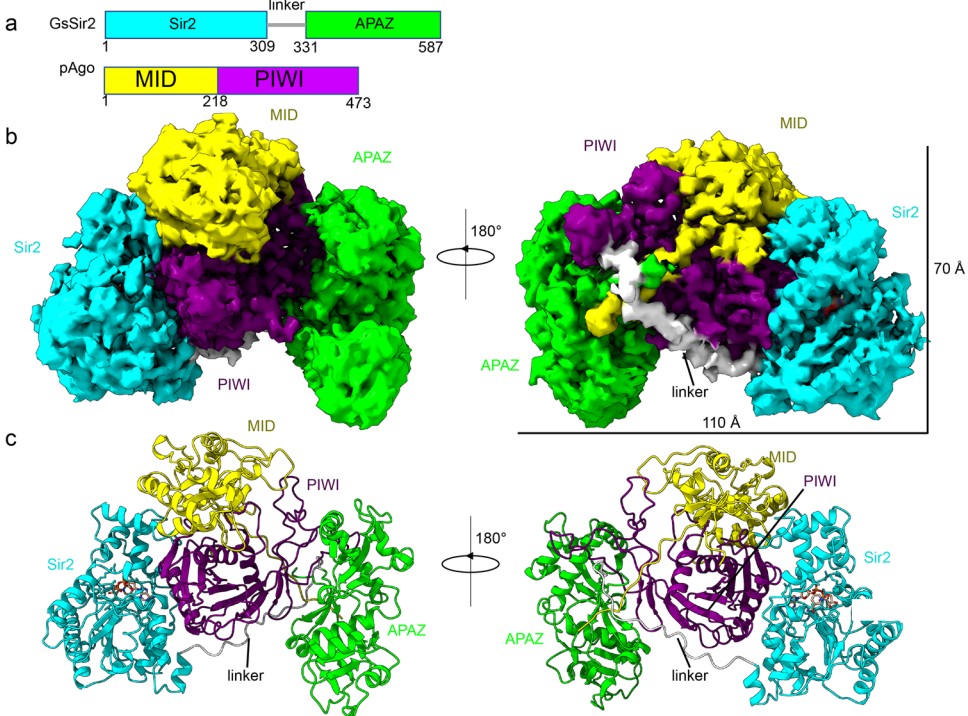

**Fig. 1 | Cryo-EM analysis of the SPARSA complex. a** Linear representation of domain organization and schematic of GsSir2 and pAgo used in this study. **b** Cryo-EM reconstruction with individual domains segmented and colored in **a. c** Cartoon representation of the SPARSA complex shown in two different orientations and colored by domains as indicated.

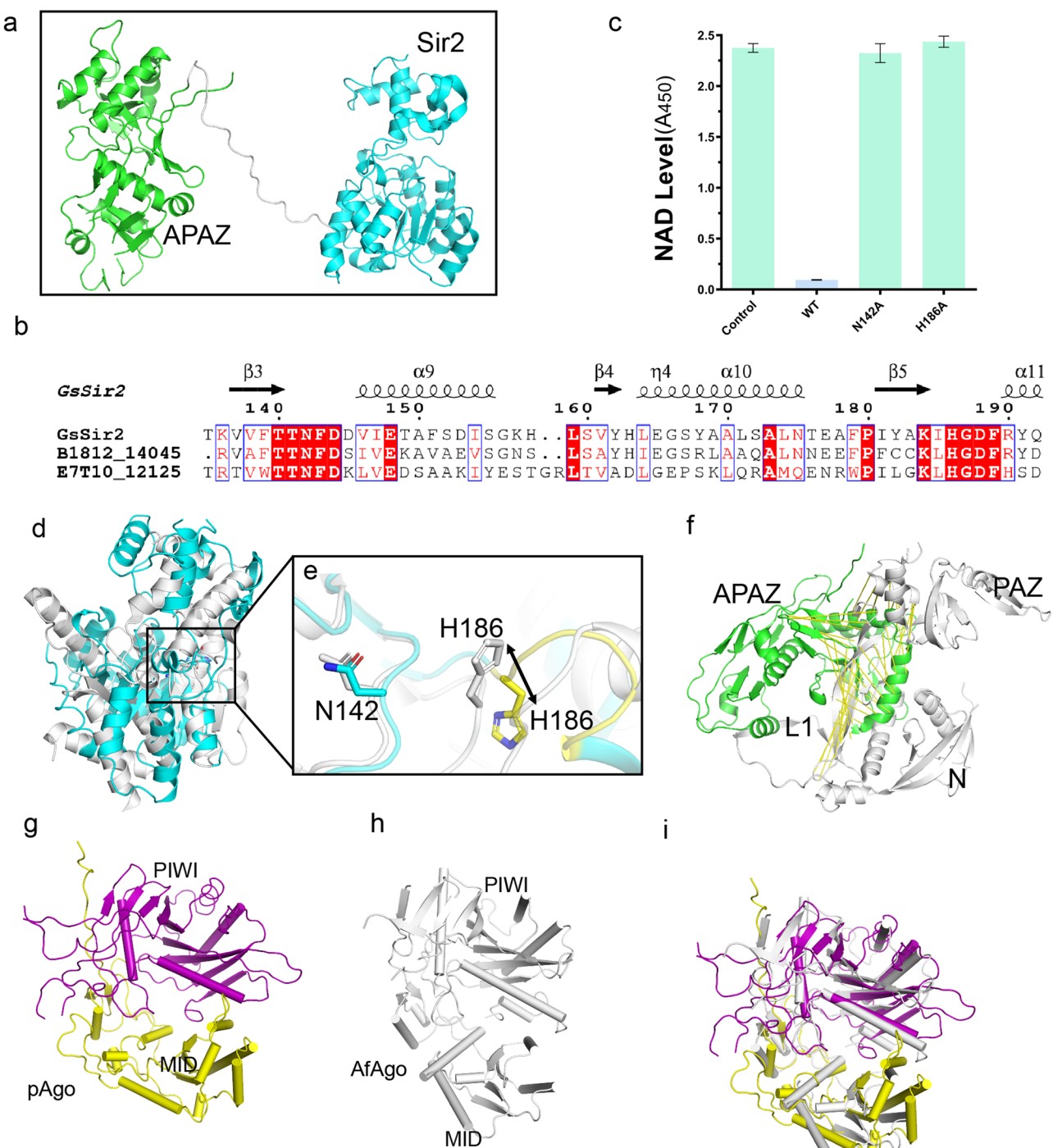

**Fig. 2 | Structural analysis of GsSir2 and pAgo. a** The structure of GsSir2 is composed of an N-terminal Sir2 and C-terminal APAZ domain, i.e., the N-terminal Sir2 domain (residues 1–309, colored cyan) and the C-terminal APAZ (residues 331–587, colored green). The linker (residues 309–331) interconnecting the two domains is shown as a gray line. **b** Sequence alignment of Sir2 and its homologs. The conserved putative catalytic residues are indicated using a circular ring in cyan. **c** NAD⁺ assumption of wild-type SPARSA and the mutants on N142 and H186 SPARSA. Each value is the mean ± SD and from more than three replicates. **d** Superimposition of the GsSir2 N-terminal domain and the Sir2 domain of ThsA. **e** The close-up view of the overlay shows the difference in the Sir2 domain, and the catalytic residues N142 and H186 of Sir2--APAZ are highlighted in the sticks. **f** Superimposition of the APAZ domain the N domain, and PAZ domain of human hAgo2 (PDB 4F3T). **g** The structure of pAgo. **h** The structure of AfAgo (PDB: 1YTU). **i** Superimposition of pAgo with AfAgo.

back of a chair. The pAgo protein is clamped tightly by the N-terminal and C-terminal Sir2 domains via extensive interaction areas with estimated buried interfaces of 1400 and 1500 Å², respectively (Fig. 1b, c).

GsSir2 appears to exhibit a Z-shaped structure and consists of two parallel domains, including the N-terminal Sirtuin-like domain (Sir2, residues 1–310) and the C-terminal APAZ domain (residues 332–583)

(Fig. 2a). A long linker (residues 311–331) with a length >45 Å was observed, bridging the two domains (Fig. 2a). Although there is low sequence identity (<10%) between typical eukaryotic and prokaryotic Sir2 (Supplementary Fig. 3a, b), the N-terminal Sir2 domain contains two subdomains, including a classic Rossmann-like fold with a six-stranded parallel β-sheet sandwiched by three helical layers, and a

small domain consisting of α helices that resembles a lid covering the Rossmann subdomain (Fig. 2a).

Based on the sequence alignment involving the N-terminal Sir2 domain and other homologous sequences, the conserved N142 and H186, which are located at the cleft between the small domain and the Rossmann domain, might be the catalytic residues of the Sir2 NADase, equivalent to the putative signature catalytic Asn-His dyad of the Sir2 NAD$^+$-dependent histone deacetylases in eukaryotes (Fig. 2b). To confirm whether these residues are the enzymatic sites, we introduced single alanine mutations at H186 and N142. Subsequently, we incubated 20 μl of 1 μM purified SPARSA or its variants with gRNA and the cognate tDNA and then tested their NADase activities, which were measured by the residual NAD$^+$ levels, by reading the absorbance at 450 nm after the reaction was terminated. Based on the results, nearly no NAD$^+$ remained in the wild-type SPARSA; in contrast, almost no reduction in the amount of NAD$^+$ was detected for either the H186A or N142A mutant, suggesting that the mutations eliminated the NADase activity of GsSir2. Taken together, these results confirmed that H186 and N142 form the catalytic dyad that participates in the hydrolysis of NAD$^+$ (Fig. 2c).

The structure of the N-terminal Sir2 domain was subjected to the Dali server (http://ekhidna2.biocenter.helsinki.fi/dali/) for a search of proteins with globally similar structures in the Protein Data Base (PDB). We found that the N-terminal Sir2 was structurally similar to ThsA, the founding member of prokaryotic Sir2 domains with NADase activity in the Theoris antiphage system[27,28] (Fig. 2d). However, the overall root-mean-square deviation (RSMD) was 5.8 Å (for Cα atoms) over 157 residues, suggesting a considerable difference in the structural aspect. The catalytic N142 can be well aligned with N153 of the autoactive ThsA (PDB 6LXH), whereas the sidechain of H186 is oriented in the opposite direction to that of ThsA (Fig. 2e), which may explain the inactive SPARSA. The differences are also reflected in the small domain of Sir2, in which the triangular structural mode protruded from the Rossmann-like fold of ThsA (Supplementary Fig. 4a).

The C-terminal APAZ domain should be analogous to the N-PAZ lobe in conventional Agos, although there was no detectable sequence identity[20]. However, no structural similarity was detected between APAZ and the N-PAZ bilobe of the Ago-clade proteins (Fig. 2f). The APAZ domain could be further divided into two subdomains, namely, APAZ-a and APAZ-b (Supplementary Fig. 5a). The APAZ-a subdomain is composed of a four-parallel β-sheet core capped by three helices, whereas the APAZ-b consists of twisted parallel β-sheets surrounded by three α helices and a β-hairpin bent towards the other lobe. The APAZ-a and APAZ-b subdomains correspond to the N and PAZ domains of Ago-clade proteins, respectively (Supplementary Fig. 5b).

### Short pAgo adopts a typical MID-PIWI fold

The structure of pAgo in the SPARSA system consists of a MID domain (residues 1–218) and a PIWI domain (residues 219–473) as predicted, which shows a typical MID-PIWI bilobed fold similar to the Ago-clade[29] and the PIWI-clade proteins[30] (Fig. 2g). In addition to the presence of N-terminal β-hairpins, the MID domain mainly shows α-helices that clamp three parallel β-sheets. The PIWI domain consists of four α-helices separated by nine parallel β-sheets (Fig. 2h). ADali search (http://ekhidna2.biocentre.helsinki.fi/dali/) analysis with the structure of pAgo as the bait revealed that pAgo shares the highest structural similarity with the Ago-clade proteins, especially with the MID-PIWI domain of *Archaeoglobus fulgidus* Ago (AfAgo)[31] with a *Z* score of 24 and RMSD of 3.1 Å, which confirms that they are similar (Fig. 2h, i).

An architecture similar to that of the typical long Agos is cooperatively formed by pAgo and the APAZ portion of GsSir2 (Supplementary Fig. 6a). A large positively charged surface is created to form the nucleic acid-binding channel (Supplementary Fig. 6b), which is similar to that of typical long Agos[29,32]. The architecture adopts an extended conformation that exhibits a wider central cleft than that of

eukaryotic Agos (Supplementary Fig. 7a–c)[33]. Additionally, the overall architecture is shallower than that of canonical Agos, such as human Ago2 (hAgo2)[33] (Supplementary Fig. 6d), due to the lack of L1 (residues 174–183), which is responsible for narrowing the channels of N-APZ and L2 that connect the N-PAZ lobe and the MID-PIWI lobe.

### The interactions between GsSir2 and pAgo

In the SPARSA heterodimeric complex, a long linker (residues 310–331) that bridges the N- and C-terminal domains of GsSir2 was successfully built in the model (Fig. 2a and Supplementary Fig. 8a). The linker densities are unambiguous, which may be influenced by the interactions with pAgo. We found that the long linker is enriched with hydrophobic residues and is accommodated on a hydrophobic surface lining the PIWI domain of pAgo close to the loop (Supplementary Fig. 8a). Several hydrophobic residues of the linker pointed to the hydrophobic pockets of pAgo, contributing to their interactions. In addition to the hydrophobic interactions, hydrogen bonds formed between the linker and pAgo (Supplementary Fig. 8b).

The N-terminal Sir2 domain engages in extensive interactions with pAgo, with a total buried interfacial area of ~1400 Å$^2$ (Supplementary Fig. 8c). Notably, this interaction predominantly relies on the PIWI domain, distinguishing it from the SPARTA, in which the MID domain of pAgo packs against the TIR. Considerable intermolecular interactions between pAgo and the C-terminal APAZ domain of GsSir2 are also observed (Supplementary Fig. 8d). Similar to SPARTA, the interface between APAZ and PIWI accounts for the largest interaction regions, mainly involving hydrogen bonds, with a buried area of ~1100 Å$^2$.

### Structural basis for the recognition of the N-terminal Sir2 domain by NAD$^+$

In the cryo-EM density of SPARSA, a density within the crevice near the catalytic residues N142 and H186 was observed, which was compatible with the NAD$^+$ molecule (Fig. 3a). As no external NAD$^+$ molecules were introduced, the bound NAD$^+$ may be an endogenous molecule that is trapped during protein purification; these results are consistent with the findings of the previous study, in which SPARSA was shown to bind NAD$^+$ in vivo[21]. Similar to the classic Sir2-like sirtuins, NAD$^+$ was bound in the cleft between the Rossmann fold and the small domain[34] (Fig. 3b). However, NAD$^+$ adopts a bent conformation that differs from the extended conformation adopted by most NAD$^+$-binding Rossmann proteins[35].

The NAD$^+$ molecule embraces α2 of the Sir2 domain, which causes the nicotinamide (NAM) ring to be almost parallel to the adenine ring. NAD$^+$ is buried with surrounding surface interfaces of ~850 Å$^2$ and engages in extensive hydrogen bonds and hydrophobic interactions. The NAM ribose moiety is positioned to the pocket formed by F143, F93 and Y89. The NH$_2$ group of NAM is held in place by forming hydrogen bonds with the sidechain of D144 and the main chain of F143. Additionally, the 2′OH of the NAM ribose interacts with the main chain of T141. The NAM ribose portion is stabilized by a hydrogen bond between its 3′OH and the sidechain of D230 (Fig. 3c). The pyrophosphate group forms hydrogen bonds with the side chains of S227, S38, and R229 and the main chain amide between A26 and G25. The adenosine moiety of NAD$^+$ contacts the sidechain of R30 and the main chain of F287. Additionally, adenosine exhibits hydrophobic contacts with the aromatic ring of Y143, L112 and L34, which also contributes to NAD$^+$ binding (Fig. 3c). Adenine ribose forms hydrogen bonds with E262 and Y226.

To assess the involvement of these residues in NAD$^+$ binding, we individually substituted them with alanine, and the purified proteins were employed in NAD$^+$ assays in vitro. The test results demonstrated that the Sir2 mutants, including R30A, S38A, R190A, S227A and E262A, exhibited a significant decrease in NAD$^+$ levels, suggesting that NADase activity was retained. Conversely, the levels of NAD$^+$ in the mutants Y226A, D144A, F143A and F287A were comparable to those in the

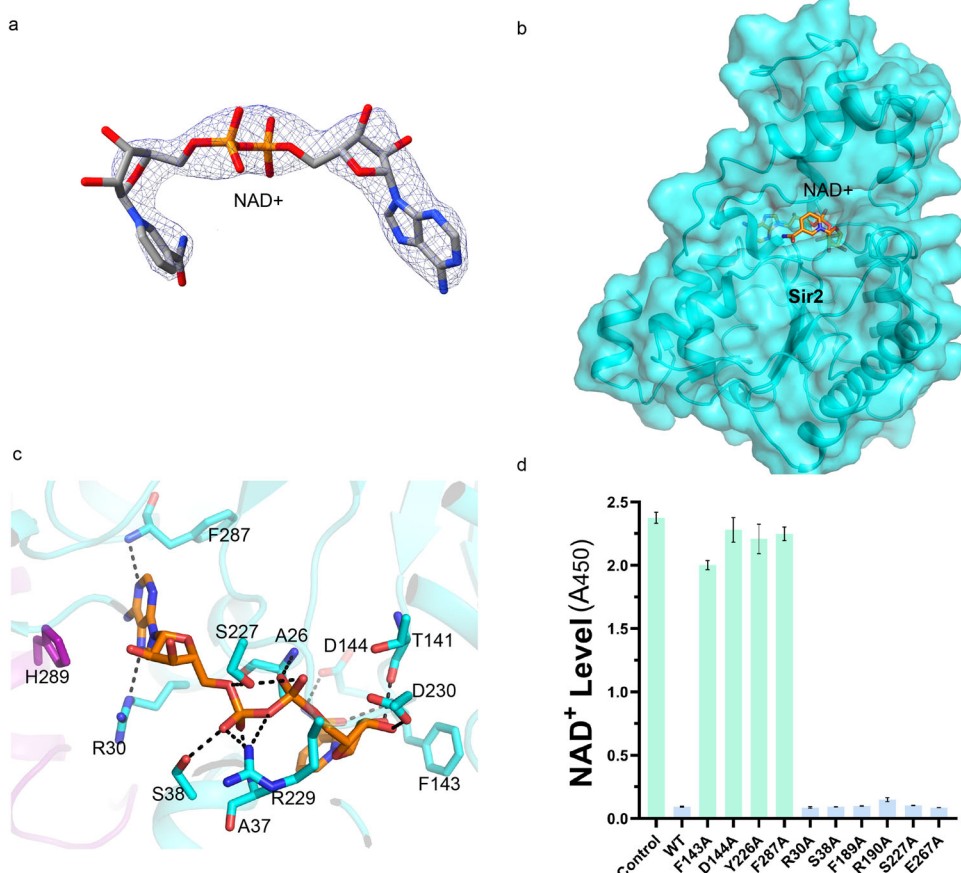

**Fig. 3 | NAD⁺ recognition mechanism of SPARSA. a** The cryo-EM density of bound NAD⁺. **b** A cartoon representation of NAD⁺-bound in Sir2. NAD⁺ is shown as sticks colored in yellow by element, and the N-terminal Sir2 domain is colored in cyan and shown in surface mode. **c** An expanded view of the NAD⁺ interaction with SPARSA. The residues involved in NAD⁺ binding are shown in sticks. **d** The in vitro NADase assays of the wild-type and the variants of SPARSA. Each value is the mean ± SD and from more than three replicates.

control group, suggesting that NADase activities were completely lost in the mutants compared to the wild-type SPARSA (Fig. 3d), which indicates the critical role of these residues in NAD⁺ binding.

**Structural basis for the recognition of gRNA**

In the SPARSA antiphage system, the recognition of tDNA by gRNA was needed for the NADase activity of the N-terminus of Sir2[21]. Accordingly, to gain insight into the mechanism underlying the NADase activation of GsSir2, we investigated the structure of SPARSA in complex with gRNA and tDNA. All three reported pAgos exhibited a preference for the gRNA with 5'-end phosphorylation[20–22] in vivo and in vitro with high affinity (nM level)[20,21]. Thus, the phosphorylated 42 nt gRNA and the complementary tDNA were selected and used to perform the cryo-EM analysis (Supplementary Fig. 9).

The cryo-EM structure of SPARSA in complex with the guide and target nucleic acid molecules was determined at a resolution of ~3.1 Å (Supplementary Table 1, Supplementary Fig. 10a–c). The heteroduplexes of gRNA and tDNA are eventually embedded in the nucleic acid-binding channel formed by pAgo and the APAZ portion of GsSir2 (Fig. 4a). The architecture of pAgo-APAZ with gRNA and tDNA more closely resembles that of the prokaryotic long Agos compared to the architecture of eukaryotic Agos (Supplementary Fig. 6d). In the PIWI domain, the catalytic residues within the DDE motif were substituted, which eliminated its nuclease activity and facilitated the formation of the SPARSA complex with gRNA and the uncleaved target strand. The gRNA and the tDNA can be monitored from positions 1–22, and the opposite 1–22', respectively (Fig. 4b).

Except for the first two nucleotides at the 5' end of the gRNA and the opposite 3' region of the tDNA, the remaining nucleotides in the gRNA are fully Waston-Crick paired with the tDNA strand (Fig. 4b), which is distinct from the Agos with slicer activity, in which only the seed region (positions 2–8 of the gRNA) is paired. The kink that disrupts the gRNA helical stacking between bases 6 and 7 induced by helix-7 is not observed in the gRNA. An unambiguous density of NAD⁺ is also observed in the SPARSA-gRNA/tDNA complex, which may be bound endogenous NAD⁺ before gRNA and tDNA are added (Supplementary Fig. 11a–c). The NADase assays show that the NADase activity of SPARSA-gRNA-tDNA is low at a low reaction temperature (Supplementary Fig. 11d). This low activity may explain why the complex of SPARSA-gRNA-tDNA remains uncleaved during the assembly process and cryo-EM data collection. Therefore, the SPARSA-gRNA-tDNA complex, bound to endogenous NAD+, is indicative of a prehydrolyzed state.

The heteroduplex is stabilized by a network of intermolecular hydrogen bond interactions, as shown in Fig. 4b. The insertion of Y359 between T1' and A1' of the 3'-region tDNA disrupts helical base stacking, splaying the terminal two bases away from the heteroduplex. A similar disruption at the 3' end of the target strand by aromatic residues was also reported in the ternary complex structures of MapAgo[36], RsAgo[37] and TtAgo[38] (Supplementary Fig. 12a–c). The first three nucleotides of the gRNA adopt an A-form-like conformation, similar to the Agos reported in previous studies[32,37,39]. The structure highlights the nucleic acid-binding conservation of the 5' end of the gRNA, which is tethered to pAgo through several interactions to form a very tight

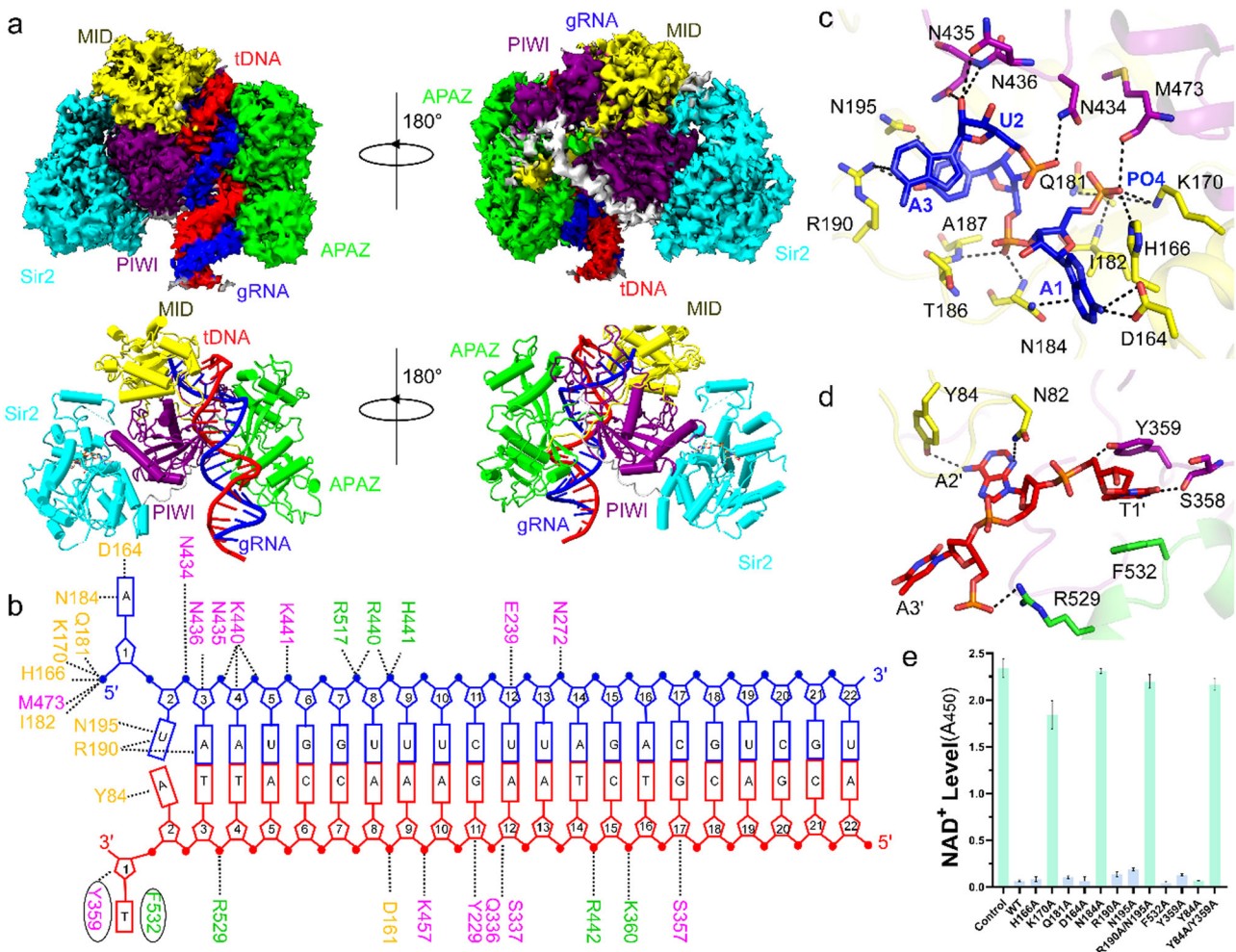

**Fig. 4 | The cryo-EM structure of SPARSA with gRNA/tDNA. a** The cryo-EM map (upper panel) and the structure of SPARSA with the gRNA/tDNA quaternary complex (bottom panel). **b** Schematic representation of the overall interactions between SPARSA and the gRNA/tDNA duplex. The bases involved in Watson–Cricks base pairing are indicated by red boxes and connecting lines. The gRNA and the tDNA are shown in the sticks and colored similar to **a**. The SPARSA are shown in cartoon mode and colored as in Fig. 1. **c** Recognition of the 5' end of the gRNA of SPARSA. **d** The interaction between pAgo and the 3'-region of tDNA. **e** In vitro assays of the wild-type SPARSA system or the mutants on the 5' end of the gRNA and the 3'-end of the tDNA. Each value is the mean ± SD and from more than three replicates.

binding pocket of the MID domain (Fig. 4c). In addition, the 5' phosphate interacts with several residues, including the side chains of H166, K70 and Q181 (Fig. 4c). The lysine and asparagine residues participate in recognition of the phosphate group, which was also observed in previously reported Agos[29,32]. These amino acid residues are conserved among the pAgo (H202 and Q222 in pAgo of SPARTA[20]), suggesting that the recognition mode of the 5' phosphate of gRNA was conserved by the pAgos.

The first two bases of the gRNA are splayed away and were bound with the MID domain, in which D164 and N184 form hydrogen bonds with 5'-A, and the side chains of R190 and N195 are directly recognized by the second U. The sidechain of R190 also contributes to the interaction with the adenine of the third A (Fig. 4c). The AUAs are the only bases on the guide strand that forms base-specific contacts with the MID domain of pAgo, which could explain why SPARSA showed bias towards the 5'-AU dinucleotides[21]. Except for intermolecular contacts that are mediated by the MID pocket, the remaining intermolecular interactions line the binding cleft. These interactions are nonspecific and involve a nucleic acid sugar-phosphate backbone that spans bases 4–12 on the guide strand (Supplementary Fig. 13a, b).

## Structural basis for the recognition of the 3'-end of tDNA

The structure shows that the binding of tDNA mainly depends on hydrophobic interactions between the 3' end of the tDNA and some residues of the MID, PIWI and APAZ domains of GsSir2 (Fig. 4d). Specifically, the thymine of T1' is sandwiched by the ring of the aromatic residue Y359 in the MID domain and F532 on the other side from the APAZ domain. The thymine and phosphate backbone of T1' also form hydrogen bonds with Y359 (Fig. 4d). The base of A2' forms a direct hydrogen bond with the side chains of Y84 and N82 in the MID domain. Along with the sequence-specific binding mediated by the first two bases, the intermolecular interactions between tDNA and SPARSA also depend on the nonspecific nucleic acid sugar-phosphate backbone that spans position 3 to 17 dispersed over the remaining part of the DNA strand (Supplementary Fig. 13c, d), which is similar to other Agos[40].

To evaluate the functional importance of these residues, which were identified for the recognition of gRNA and tDNA, the above-mentioned residues were substituted, and the resulting mutants were tested by in vitro NADase assays. The residues were mutated to bind 5'-phosphate, and single mutations H166A and Q181A did not affect NADase activity. In contrast, mutating K170 considerably decreased

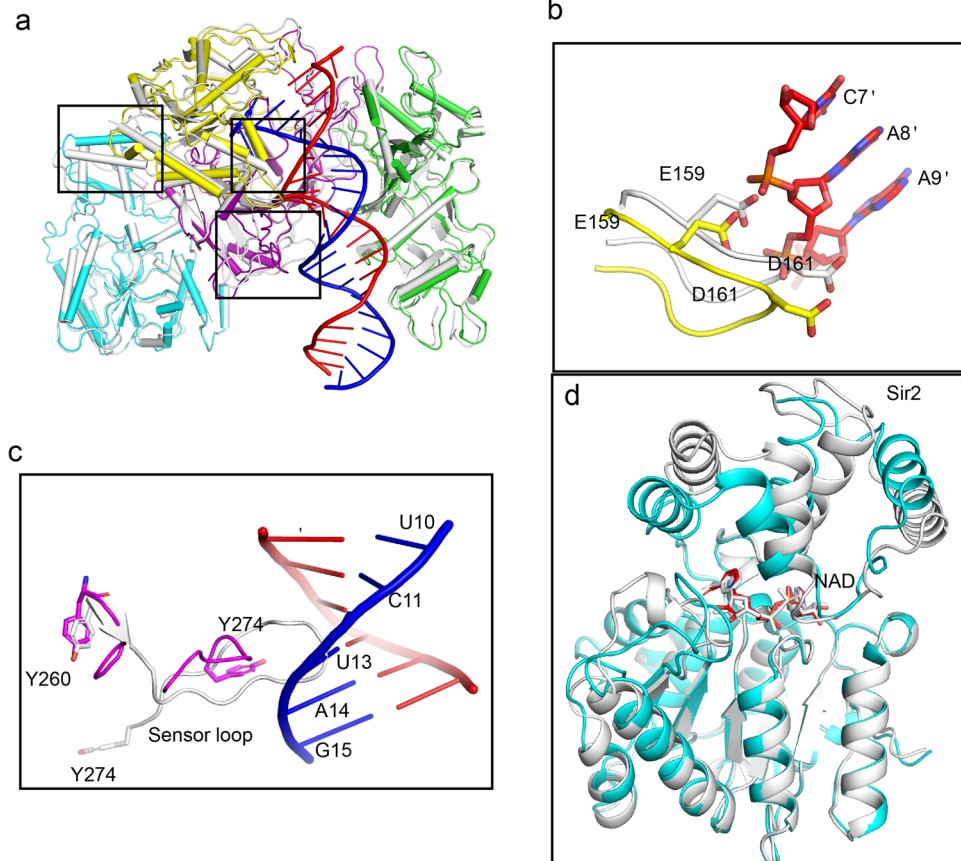

**Fig. 5 | Structural conformation change induced upon the binding of nucleic acids. a** Overlay of SPARSA and SPARSA-gRNA/tDNA/NAD⁺ in cartoon representation; apo SPARSA is shown in white, and SPARSA-gRNA/tDNA is shown in Fig. 4. **b**, **c** Close-up view of the conformational changes in pAgo of SPARSA with and without gRNA and tDNA. **d** Comparison with SPARSA and SPARSA-RNA/DNA reveals the conformational change of the Sir2 N-terminal domain induced upon the binding of RNA and target. The Sir2 N-terminal domains are shown in cyan and grey in the cryo-EM, respectively, and the bound NAD⁺ is shown as sticks.

the NADase activity of GsSir2. When the residues involved in the 5′-AUA binding of gRNA were mutated, N184A abolished its NADase activity, whereas a single mutation on other residues did not influence its NADase activity. However, the R190A/N195A double mutant also showed no NADase activity. Similarly, single mutations in the residues needed for target binding maintained NADase activity, and in contrast, when Y84 and Y359 were simultaneously mutated, NADase activity was completely lost (Fig. 4e). Taken together, these results confirmed the key residues of SPARSA for nucleic acid-binding.

**The sensor loop in the PIWI domain that monitors the formation of gRNA-tDNA also exists in SPARSA**
The overall structures of SPARSA and the gRNA/tDNA-bound SPARSA are similar (0.738 Å) (Fig. 5a). In SPARSA, the channel between the PAZ and MID domains does not open after gRNA/tDNA binds, which is different from other Agos[39,41,42].

Only subtle local structural rearrangements are mainly observed within pAgon in the regions that are close to the bound gRNA and the tDNA. The loop in the MID domain (residues 76–82) is bent towards tDNA to interact with the 3′ end of the tDNA strand, and anther loop (residues 157–164) of the MID domain is bent away from the nucleic acids to avoid steric clashes with the tDNA duplex (position 7′–9′) (Fig. 5b). A long loop (residues 260–274) protruding from the PIWI domain bends away from the gRNA-tDNA to avoid steric clashes, which is the sensor loop that acts as a sensor to detect the formation of the gRNA-tDNA[43] (Fig. 5c). In addition, conformational changes in the α helices of the small domain in Sir2 are observed (Fig. 5d), which was also observed in the Sir2 domain of ThsA[27]. There is little structural

change in the Rossmann domain, especially among the residues involved in NAD⁺ binding. The bound NAD⁺ molecule adopted nearly the same conformation in our two SARSA structures (Fig. 5d).

**Putative mechanism of the activation of Sir2 NADase**
The inactive SPARSA is capable of binding NAD⁺, suggesting that its activation does not involve the exposure of the active site for NAD⁺, which differs from ThsA[44]. To further decipher the molecular mechanism underlying the activation of Sir2 NADase by the binding of gRNA/tDNA, we investigated the postcatalytic structure of SPARSA-gRNA/tDNA in the presence of NAM and ADPR. NAD⁺ was added to the purified SPARSA-gRNA/tDNA with different molar ratios. However, the addition of NAD⁺ decreased the stability of the N-terminal Sir2 domain. Despite extensive attempts, we failed to obtain the structure of SPARSA-gRNA-tDNA with the hydrolysis production of NAD⁺.

To investigate the catalytic mechanism of GsSir2, we conducted 1000-ns-long molecular dynamics (MD) simulations on the structures of SPARSA-NAD⁺ and SPARSA-gRNA/tDNA/NAD⁺ in an aqueous solution. The convergence of the MD simulations was established by calculating the root-mean-square deviations (RMSDs) of the backbone atoms of SPARSA and SPARSA-gRNA/tDNA as well as those of the heavy atoms of NAD⁺ (Supplementary Fig. 14a–d). The most representative structures of each complex were obtained through cluster analysis of the last 200-ns-long MD trajectories (Fig. 6a). Although the binding modes of NAD⁺ in the most representative MD structures closely resemble those observed in the cryo-EM structures, two water molecules (hereinafter water I and II) are specifically found to surround the

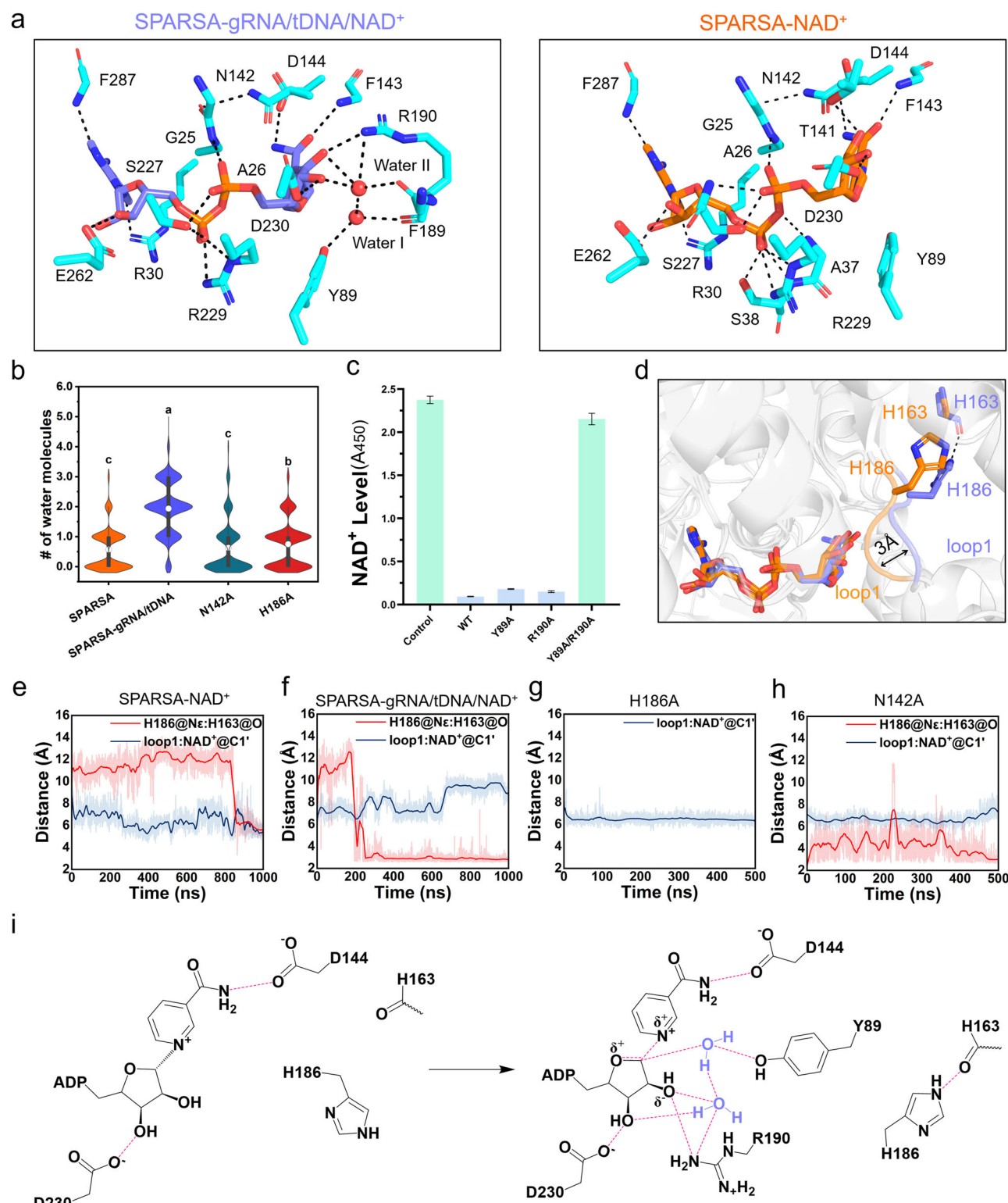

NAM ribose moiety in SPARSA-gRNA/tDNA/NAD+. The water mobility analysis suggests that the average number of water molecules in SPARSA-gRNA/tDNA/NAD+ ($1.9 \pm 0.9$) is threefold higher than that in SPARSA-NAD+ ($0.6 \pm 0.6$) across the MD simulations (Fig. 6b). Approximately two water molecules were found in SPARSA-gRNA/tDNA/NAD+ during the final 200-ns MD simulation (Fig. 6b and Supplementary Fig. 15a–d). Water I and water II form many hydrogen bonds with the NAM ribose moiety of NAD+, the sidechain of Y89, the backbone of F189, and the sidechain and backbone of R190. Notably,

this hydrogen bond network stabilizes the water molecules in positions close to the anomeric carbon (C1′) of NAD+ with atomic distances of $3.2 \pm 0.2$ Å and $3.4 \pm 0.3$ Å, respectively. As previously reported, the cleavage of NAD+ in fungal NADase involves a nucleophilic attack on C1′ originating from the α-face by a water molecule[45]. Therefore, the similar positioning of water molecules in the Sir2 domain provides possibilities for an α-face nucleophilic attack on C1′, leading to the hydrolysis of NAD+. Corresponding to this speculation, the double-point mutations of Y89A and R190A abolish the NADase activity of

**Fig. 6 | MD simulations revealed the NAD⁺ hydrolysis mechanism of the Sir2 domain of GsSir2. a** The most representative structures are SPARSA-NAD⁺ and SPARSA-gRNA/tDNA/NAD⁺. Sir2, APAZ, PIWI, MID, and gRNA/tDNA are represented by cyan, green, magenta, yellow and orange cartoons, respectively. NAD⁺ is colored in orange (SPARSA) or purple (SPARSA-gRNA/tDNA). NAD⁺ and key residues are shown in sticks. The water molecules are represented by red spheres. **b** The number of water molecules around the C1′ atom of NAD⁺ within a cutoff of 3.5 Å. Data are represented as violin plots with box and whisker (lower and upper hinges correspond to the first and third quartiles, mean values are presented by circle within the box, whiskers extend from the hinge to the largest/smallest value). The maximal, central, and minimal values observed are as follows: For SPARSA, the values were 3, 0, and 0; for SPARSA-gRNA/tDNA, they were 5, 2, and 0; for N142A, the values were

5, 0, and 0; and for H186A, they were 4, 1, and 0. The statistical significance was analyzed by using one-way ANOVA followed by Bartlett test. A $p$ value of less than 0.05 was considered statistically significant. **c** The NADase activity of WT SPARSA-gRNA/tDNA and the Y89A, R190A and Y89A & R190A SPARSA-gRNA/tDNA mutations. Each value is the mean ± SD and from more than three replicates. **d** A remarkable conformation of loop1 ($^{186}$HGD$^{188}$) caused by the hydrogen bond interaction between H186 and H163. **e, f** The distance between the Nε atom of H186 and the O atom of H186 and the distance between loop1 and C1′ in SPARSA-NAD⁺ (**e**), SPARSA-gRNA/tDNA/NAD⁺ (**f**), H186A (**g**) and N142A (**h**). **i** Schematic of the NAD⁺ hydrolysis mechanism of the Sir2 domain. The hydrogen bond interaction between H186 and H163 may allow the presence of water molecules for an α-face nucleophilic attack on C1′.

GsSir2 (Fig. 6c), which confirms the essential role of water molecules in NAD⁺ hydrolysis.

Although the proposed mechanism eliminates the direct involvement of the Asn-His dyad (N142 and H186) in the catalysis of NADase, they might play a key role in regulating the presence of water molecules at the catalytic site. Structural comparison between the MD representative structures of SPARSA-NAD⁺ and SPARSA-gRNA/tDNA/NAD⁺ revealed a remarkable conformational change in the loop (residues H186-D188, namely, loop1) close to the NAM ribose moiety of NAD⁺, i.e., loop1 moved outwards from the NAM ribose moiety for ~3 Å when gRNA/tDNA bound (Fig. 6d, Supplementary Fig. 16a, b). The movement results in the opening of a channel for water molecules to enter the catalytic site for nucleophilic attack on the C1′ of NAD⁺. Additionally, a hydrogen bond formed between the sidechain of H186 from loop1 and the backbone of H163 is observed only in SPARSA-gRNA/tDNA; therefore, this hydrogen bond might be connected to the movement of loop1. Calculations of backbone root-mean-square fluctuation (RMSF) of the residues in loop1 revealed that the presence of a hydrogen bond between H186 and H163 contributes to the enhanced stability of loop1 (Supplementary Fig. 16c, d). Further analysis involving the movement (distance between the centre-of-mass of loop1 and the C1′ atom) and hydrogen bond (distance between the Nε atom of H186 and the O atom of H163) during the MD simulations showed that the movement of loop1 was induced by the formation of the hydrogen bond between H186 and H163 (Fig. 6e, f). The movement is abolished in mutagenesis of H186A, which further supports the previous finding (Fig. 6g, Supplementary Fig. 16b). This also occurred in the N142A mutant, in which the eliminated hydrogen bond between N142 and G25 led to a displacement of the sidechain of F143. The ensuing steric hindrance causes H186 to move towards C1′, subsequently destabilizing the hydrogen bond between H186 and H163 to restrict the movement of loop1. (Fig. 6h, Supplementary Fig. 16c). As a result, the average number of water molecules surrounding the NAM ribose moiety decreased significantly in H186A and N142A compared to the WT of SPARSA-gRNA/tDNA (Fig. 6b). Similar results are found in replica MD simulations with different initial velocity distributions (Supplementary Fig. 17). As water molecules play a key role in catalysis, these results might explain why mutation of the Asn-His dyad inactivates the NAD⁺ hydrolysis of GsSir2 (Fig. 2c).

Collectively, based on the MD simulations, we propose that the movement of loop1 triggered by the hydrogen bond interaction between H186 and H163 probably allows the water molecules to perform a nucleophilic attack on C1′, leading to the cleavage of NAD⁺ in Sir2 NADase (Fig. 6i).

## Discussion

In the evolutionary arms race against phages, bacteria have evolved a myriad of antiviral defense systems, which involve different strategies, such as restriction-modification (RM) systems and abortive infection. Depletion of the essential signaling molecules that cause host death or growth arrest to clear infected cells is a common strategy employed by bacteria to combat phage infection[46]. Many bacteria deploy the rapid

hydrolysis of NAD⁺ to promote their virulence or combat phage infection. Recently, studies on short pAgos complexes from different clades have revealed that these systems function in phage defense via abortive infection[20–22]. In the SPARTA and SPARSA antiphage systems, NAD⁺ is depleted by the associated effector domain, Sir2 or TIR. The mechanism of SPARTA NADase activation has been reported recently and depends on the tetramerization of the TIR domain mediated by the MID:MID interaction of pAgo[47–50]. Tetramerization of SPARTA, which is induced by the RNA/DNA duplex, promotes the formation of the substrate binding pocket mediated by the BB loop and EE surface, thereby triggering the SPARTA complex's NADase activity[43,47,49,51,52].

Sir2 domain proteins are broadly distributed in eukaryotes and prokaryotes. In eukaryotes, these proteins act as NAD⁺-dependent deacetylases or ADP ribosyltransferases[53]. Recent findings have revealed that Sir2 domain-containing proteins can function as NAD⁺ hydrolases[21,26,27,54]. The NADase activity of ThsA is activated by 3′ cADPR, which binds to the SLOG domain of ThsA. Structural superimpositions of ThsA from *B. cereus MSX-D12* (BcThsA) and *Streptococcus equi* (SeThsA) reveal that the N-terminal α3 helix in the small domain of Sir2 participates in its tetramerization, which covers the active site region. The binding of 3′ cADPR induces conformational changes to destabilize its oligomerization, enabling NAD⁺ to the active sites[27,28]. However, the activation of SPARSA NADase does not involve exposure of the active site for NAD⁺ hydrolysis, and apo SPARSA can bind NAD⁺. Although structural changes in the small domain of the N-terminal Sir2 are observed, the mechanism of activation should be similar to that of ThsA due to the lack of the structure of full-length SPARSA, especially NAD⁺-bound SPARSA[43].

In addition to the conformational changes in the small domain of the N-terminal Sir2, the orientation of the loop containing H186 shifts away from NAD⁺ in the SPARSA-gRNA/tDNA. The molecular dynamics simulations revealed that the shifting away of NAD⁺ results in the formation of a space to accumulate two water molecules, which are mainly stabilized by Y89 and R190. Water molecules should undergo a nucleophilic attack on C1′ of NAD⁺ to activate NADase, which is similar to other NADases[45].

## Methods
### Protein purification

The genes encoding GsSir2 (NP_952413.1) and pAgo (NP_952414.1) were synthesized cloned, and inserted into the pET21a (+) vector to express GsSir2- and pAgo, respectively, to assemble the SPARSA complex. However, no soluble recombinant proteins were obtained. Thus, they were constructed into a modified pET21a (+) vector, GsSir2 and pAgo, which were coexpressed, and a sequence for the ribosome binding site (AGGGAA) was placed upstream of the AUG codon of pAgo to coexpress a 6x His-SUMO-SPARSA complex. Then, the SUMO tag was removed by TEV. Then, Sir2-RBS-pAgo was cloned and inserted into the pET21a (+) vector containing an N-terminal 6X His-tag to obtain the 6x His-SPARSA heterodimeric complex[55].

The corresponding recombinant plasmids were transformed into *E. coli* BL21(DE3) cells and cultivated in LB medium at 37 °C in

the presence of 100 μM Amp until the $OD_{600}$ reached 0.6. Subsequently, the temperature was decreased to 16 °C, and the protein was expressed for 16 h by adding IPTG to a final concentration of 0.4 mM. Next, the cells were pelleted by centrifugation and resuspended in buffer containing 50 mM Tris-HCl, pH 8.0, and 150 mM NaCl. The cells were then lysed by ultrasonication, and the lysate was centrifuged at $17,000 \times g$ and 4 °C for 30 min. The supernatant was then loaded onto a $Ni^{2+}$-NTA column (Qiagen) for purification of our target recombinant proteins. After the supernatant was washed with 50 mM imidazole with 50 mM Tris-HCl, pH 8.0, 150 mM NaCl, the target proteins were eluted with buffer with 50 mM Tris-HCl, pH 8.0, 150 mM NaCl and 250 mM imidazole. Fractions containing the target protein were pooled and concentrated to 0.5 mL, which was purified with a Superdex 200 increase column (Cytiva Life Sciences) equilibrated with buffer containing 20 mM Tris-HCl, pH 8.0, and 150 mM NaCl.

### In vitro assembly of the SPARSA-gRNA-tDNA complex

To assemble the SPARSA-gRNA-tDNA complex, the purified SPARSA sample was incubated with 42 nt 5′ phosphorylated gRNA[56] at a molar ratio of 1:1.2. After incubation at 4 °C for 30 min, complementary tDNA[56] was added at a molar ratio of 1:1.2 and incubated at 4 °C for 30 min. The resulting sample was then subjected to a Superdex 200 Increase column (Cytiva Life Sciences). The fractions were used to perform cryo-EM experiments.

### Cryo-EM sample preparation, data collection and image processing

A 3 μL sample of the GsSir/Ago complex in the absence or presence of the DNA/RNA duplex at a concentration of 2 mg/mL was applied to the glow-discharged grid (Quantifoil Cu R1.2/1.3, 300 mesh) and plunge-frozen in liquid ethane using a ThermoFisher Vitrobot with settings of 3 sec blot-time, level 3 blot-force, and 100% humidity at 8 °C. Cryo-EM datasets were collected in multiple sessions using the EPU automated software (Thermo Fisher) on a Thermo Fisher 300 kV TEM Titan Krios G4 equipped with a Gatan K3 direct electron detector. A total of 8221 and 13058 micrographs were recorded as movie stacks in super-resolution mode at ×105,000 magnification (resulting in a calibrated pixel size of 0.42 Å/pixel) for the GsSir/Ago and GsSir/Ago/RNA/DNA complexes, respectively. For each micrograph, 40 frames were collected with a total electron dosage of $50e^-/Å^2$.

Subsequent image processing was performed using Cryosparc v4.0.1[57]. All recorded movies were subjected to patch motion correction and binned to a pixel size of 0.84 Å/pixel for further data processing. Dose-weighted micrographs were used for CTF estimation using Patch CTF[58]. Particles were chosen using the pretrained model of Topaz automatic picking[59], followed by several rounds of 2D classification to remove undesirable particles. The good particles were used for ab initio reconstruction and heterogeneous refinement, leading to exclusion of the noncomplexed particles. The relatively homogeneous particles were then selected after 3D classification and subjected to nonuniform refinement of the 3D reconstruction. According to the gold standard and using the Fourier shell correlation of 0.143, the overall resolutions are 3.6 Å for the SPARSA complex and 3.1 Å for the SPARSA/RNA/DNA complex.

### Cryo-EM model building, refinement and validation

The backbone structures were traced in the cryo-EM maps using Deeptracer[60], which guided the fitting of the Alfafold2 predicted structures into the density map. The model adjustment and building were then accomplished in Coot[61] and iteratively refined in real space using PHENIX[62]. The final structures were validated using MolProbity in PHENEX. The figures of all structures were prepared using PyMOL (Schrödinger, LLC) or UCSF ChimeraX[63].

### In vitro NADase assays

The in vitro NADase assays were carried out as previously described[20]. The gRNA and the complementary tDNA were stocked in 1 mM for use. Briefly, a reaction mixture of 20 was prepared, containing a final concentration of 1 μM purified SPARSA complex, 10 μM $NAD^+$ in buffer containing 10 mM HEPES, pH 7.5 and 125 mM KCl. Subsequently, 2 μM gRNA was added to the reaction system, which was incubated at 37 °C for 15 min. Complementary tDNA was then added and incubated at 37 °C for an additional 15 min at the same concentration of gRNA. The NAD/NADH quantitation kit (Sigma Aldrich, MAK307) was used to determine the levels of $NAD^+$ in each sample, according to the instructions provided by the manufacturer. All in vitro NADase assays were performed at least three times.

### MD simulations

To explore the potential catalytic mechanism of GsSir2, we performed atomistic MD simulations using the Amber 16 package[45]. The initial structures for the MD simulations of SPARSA-$NAD^+$ and SPARSA-gRNA/tDNA/$NAD^+$ were based on the corresponding cryo-EM structures solved in this study. The structural information on the missing segments in SPARSA was repaired using the Modeller 9v9 package[64]. Both systems were inserted into a 0.15 M NaCl water box with edge lengths of 110 Å, 110 Å, and 110 Å. The protonation states of the residues were assigned according to the corresponding $pK_a$ values calculated using the H++ webserver[65]. To counterbalance the charge of SPARSA-$NAD^+$ and SPARSA-gRNA/tDNA/$NAD^+$, two chloride ions and 39 sodium ions were added, respectively. The AMBER-ff14SB[66], AMBER-RNA-OL3[67] and AMBER-DNA-OL15[68] force fields were used to describe SPARSA, gRNA and tDNA, respectively. The parameters for $NAD^+$ were obtained from the Bryce Group[69], the TIP3P model[70], and the Åqvist potential[71], which were used for water molecules and ions, respectively. All bonds were constrained by the LINCS algorithm[72]. Periodic boundary conditions were applied. Electrostatic interactions were calculated using the Particle Mesh-Ewald (PME) method[73]. Coulomb and van der Waals interactions were truncated at 10 Å. The systems first underwent 5000 steps of steepest-descent energy minimization with 500 kcal mol$^{-1}$ Å$^{-2}$ harmonic position restraints on the complex, followed by 5000 steps of steepest-descent and 5000 steps of conjugate-gradient minimization without restraints. Then, the systems were gradually heated from 0 K up to 310 K in 50 ps in the NVT ensemble. After that, 1000-ns MD long simulations were carried out in the NPT ensemble. Temperature and pressure controls were achieved by a Nosé–Hoover thermostat and Berendsen barostat with a frequency of 2.0 ps, respectively[70,74]. The density-based spatial clustering of applications with noise (DBSCAN) method, employing a min-points value of 20 and an epsilon value of 1.7, was utilized for cluster analysis on the snapshots of final 200-ns-long MD simulations to identify the most representative structure of each system. The most representative structure of SPARSA-gRNA/tDNA/$NAD^+$ was used as the initial structure for the MD simulations of the single point mutations of N142A and H186A. Using the same setup and procedure, 500-ns-long MD simulations were carried out on each mutated complex. The number of water molecules in the catalytic site was determined by calculating the number of water molecule oxygen atoms with O atoms at a distance <3.5 Å from the C1′ atom of $NAD^+$. Hydrogen bonds were defined as present if the atomic distance between the acceptor and donor atoms was below 3.5 Å and the angle among the hydrogen-donor-acceptor atoms was below 30 degrees.

### Site-directed mutagenesis

A vector encoding wild-type SPARSA was used as the template, and oligonucleotides containing the desired mutations were designed (Supplementary Table 2). PCRs were carried out to introduce site-directed mutagenesis. All constructs were verified by DNA sequencing.

**Reporting summary**

Further information on research design is available in the Nature Portfolio Reporting Summary linked to this article.

## Data availability

Cryo-EM maps of this study have been deposited in the Electron Microscope Data Bank (EMDB) with the accession codes EMD-36384 and EMD-36385. The atomic coordinates and structure factors for the structures determined in this study have been deposited in the Protein Data Bank under the accession codes 8JKZ and 8JL0. Additional atomic coordinates referred to within this study are in the PDB under the accession codes 4F3T (human Ago2), 1YTU (*A. fulgidus* Piwi protein), 6LHX (ThsA), 5UX0 (*Mp*Ago), and 5AWH (*Rs*Ago). The data supporting the findings of this study are available from the corresponding authors upon request. Source data for the figures and supplementary figures are provided as a Source Data file. Source data are provided in this paper.

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

## Acknowledgements

This work was supported by the National Key Research and Development Program of Chain (2021YFC2301403 to S. O., 2021YFA0909500 and 2018YFA0900400 to F. L.), Nature Science Foundation of China grants (82225028, 82172287 to S.O. 321700145 to X.Z., and 22173020 to J. L.), and the Nature Science Foundation of Fujian Province (2023J06026 to X.Z). We thank Dr. Danyang Li, Yi Zeng and all members of the Cryo-EM center and the core facility of Wuhan University.

## Author contributions

X.Z. and S.O. conceived the project and designed the experiments. L.Y., Z.H. and S.Y. prepared the sample. X.X. and Y.W. collected the EM data. F.L. analyzed and calculated the EM map and built and regained the atomic model. S.X. and J.L. performed the molecular dynamics simulations. F.L., S.O., J.L., and the X.Z. discussed and analyzed the results. X.Z. and S.O. wrote the manuscripts with support from all the authors.

## Competing interests

The authors declare no competing interests.
