## [Peer Review File · Nature Communications]

Structural basis of antiphage immunity by the prokaryotic Argonaute-associated SPARSA systemREVIEWER COMMENTS

Reviewer #1 (Remarks to the Author):

In this manuscript, Zhen et al solved the cryo-EM structure of SPARSA complexes at different states and then performed the structure-guided mutagenesis and MD analysis to elucidate the molecular basis for target DNA binding and the recognition and hydrolysis of substrate NAD. Another recent study also reported the cryo-EM structure of SPARSA complex (PMID: 37311833), however, the authors first time revealed the full-length structures of SPARSA, which is a value addition to the understanding of the anti-phage systems. Importantly, the authors further revealed the catalytic mechanism of NADase in the SPARSA complex. Overall, this study is an excellent work representing a major advance in our understanding of NADase activity in anti-phage systems. This work should be of broad interest to the microbiology and biochemistry communities. I recommend its publication in Nature Communications.

I have some minor concerns below.

Comments:

1. a brief introduction to the SPARTA system should be included in the abstract section
2. The manuscript should be further proofread before publication. Typo and grammar errors should be corrected, e.g. line 46 use should be uses; line 54 eluciate should be elucidate; line 121 assembly should be assemble; line 137 1400 and 1500 should be 1,400 and 1,500
3. I recommend that the authors move some experiment details to the method section to simplify the manuscript, e.g., lines 120-127 and 262-264.
4. line 130 3.6 Angstrom may not be claimed as high resolution.
5. line 140 "Sir2 and APAZ domain, i.e.," should be deleted
6. Interestingly, another recent study suggests that the Sir2 domain would be flexible after activation, which contrasts with this work. How does the author explain the differences? In my opinion, the flexible Sir2 domain may represent another state of the catalytic cycle.
7. The NAD density was also observed in the target DNA-bound state. Perhaps consider explaining why the NAD is not hydrolyzed by the activated SPARSA complex.
8. The lid region of the SIR2 domain, for example, in ThsA and SIR2-APAZ protein (PMID: 6048923;37311833), is proposed to act as a switch to regulate the NADase activity. The lid region (aa 60-110) in the provided PDB also undergoes a conformational change with high flexibility after the target DNA binding, which is consistent with the previous observations. This point should be mentioned and discussed.
9. The NAD density in the target DNA-bound state should be shown. The target DNA-bound structure in this study may represent a post-substrate-bound state, whereas the missing sir2 domain in another recent study may represent a pre-substrate-bound state. It may be beneficial to thoroughly compare and include this information in the manuscript.

Reviewer #2 (Remarks to the Author):

The manuscript by Zhen et al. reports the cryo-EM structure of the SPARSA system, recently demonstrated to be part of defence mechanisms against phage infections in prokaryotes, triggering NAD⁺ depletion upon recognition of invader DNA. The current work elucidates the structure of the SPARSA system in its apo form and in complex with nucleic acids, providing a structural basis for understanding how tDNA binding activates the NADase activity. Key residues are identified by analysing NADase activity of a series of single mutants. Additionally, it employs molecular dynamics simulations to provide insight into the mechanism of NAD⁺ hydrolysis. As such, I believe the authors' work represent a significant contribution to understanding antiphage defence systems in bacteria and the role played by short pAgos. However, before publication several major and minor points should be addressed to increase the confidence in authors' conclusions and clarity of the manuscript.

- i. Please add a brief one-sentence explanation of Dali searches. If I understand correctly, they were used for comparative structural analysis against the PDB, but this is not clear from the text. Additionally, a supplementary table summarizing the output from the Dali server should be included (e.g. top 5 nearest neighbours).
- ii. On a related note, the authors identify ThsA as structurally similar to Sir2, however with a relatively high overall RMSD. It would be interesting to compare the Rossmann-like and small domain in isolation to ThsA. Further, the authors note the distances between N142 and H186 and their ThsA analogues differ. I would be curious to know the exact difference since it is hard to judge from Fig. 2e. Perhaps providing a figure with a more local overlap would be more informative.
- iii. Authors prepared several single mutants that abolished NADase activity. However, whether this is due to the interference with NAD⁺, gRNA or tDNA binding or structural changes within the SPARSA heterodimer is not addressed. If possible, it would be interesting to see analytical SEC chromatograms of SPARSA mutants in comparison to that of the wild type system and binding assays, particularly in the case of N142A and H186A, where MD simulations indicate they are not directly involved in catalysis as suggested earlier in the paper, and double mutant Y89A/R190A.
- iv. Based on molecular dynamics simulations authors suggest NAD⁺ is cleaved by water molecules that enter the catalytic site after a conformational change in loop1 upon gRNA/tDNA binding. This appears connected to His186-His163 hydrogen bonding. The outcomes of MD simulations are appealing, however performing replica simulations would help to increase the reliability of their findings regarding the loop movement. Additionally, it would be interesting to see the position of the loop1 before MD simulations in SPARSA-NAD⁺ and SPARSA-gRNA/tDNA/NAD⁺ as in panel Fig. 7e, and after MD for N142A and H186A mutants, as well as RMSF of the loop residues for the different systems. Finally, the authors should include a short comment on the mobility of water molecules I and II and how N142A destabilizes the His186-His163 hydrogen bond.
- v. Please add details on cluster analysis and how the number of water molecules surrounding NAD⁺ was calculated in the methods section.
- vi. Caption to Fig. 7 does not match the figure.
- vii. Fig. 6c: Please add labels indicating the location of the loop as done in Fig. 6b.
- viii. Finally, the manuscript would benefit from a couple additional rounds of refinement to improve clarity and readability. Certain concepts or structural regions are introduced late in the text or poorly defined (e.g., $\alpha 7$, $\alpha 15$, $\alpha 16$ within the Rossmann domain, $\alpha 11$, $\alpha 12$ within the PIWI domain). I would suggest they be included in the section discussing Fig. 1 along with a figure denoting regions of interest ($\alpha 7$, $\alpha 15$, $\alpha 16$ etc.).

Reviewer #3 (Remarks to the Author):

X. Zhen et al present structural and biochemical findings aiming to elucidate the mechanism of a *Geobacter sulfurreducens* SPARSA system, consisting of two domains of a short pAgo-family protein, Sir2 NADase and APAZ domains. The theme of SPARSA and SPARTA systems that initiate anti-phage

responses via hydrolysis of NAD⁺ is fascinating. The manuscript describes cryo-EM, molecular dynamics and mutational/biochemical data that could shed light on the mechanism of Gs SPARSA activation. Unfortunately, the manuscript in its current form does not achieve the goal of clarifying the mechanism. My regarding the presented data and their mechanistic interpretation are listed below. Furthermore, the manuscript is hard to read due to typographical and stylistic errors. I do not have the capacity to list such numerous errors: there are more than a dozen on each page. The authors must correct the conceptual and grammatical issues to present their findings in a clear way.

1. The major concern is whether the presented cryo-EM structures represent a physiologically relevant complex. If the complex of SPARSA with the RNA*DNA duplex activates NAD⁺ cleavage by Sir2, why does the active site of the SPARSA*RNA*DNA complex contains the substrate, rather than the product(s) or a vacant (post-release) site?
2. The authors describe many interactions in detail, and the comparison of structures with and without the duplex allows them to reveal larger conformational changes. These descriptions are not very helpful if one tries to understand the mechanism of SPARSA activation. It is unclear, how the local interactions are responsible for the large conformational changes upon RNA*DNA binding. Furthermore, are the large conformational changes in Sir2 important for catalysis? (also, what does this non-quantitative description imply: "huge conformational changes of Sir2 domain"?)
3. The abstract contains an exaggerated claim that "functions of prokaryotic counterparts are largely unknown". Researchers studying pAgo might disagree. A more careful statement could be "functions ... are less well understood".
4. In the introduction, the authors incorrectly claim that eukaryotic Ago proteins cleave DNA.
5. What is the "5'-phosphorylate"?
6. What does the statement "expression of GsSir2 and short pAgo alone in E. coli led to inclusion" mean?
7. Explain: "The pAgo was clamped tightly by the N-terminal and C-terminal domains of GsSir2 with 1400 and 1500 Å², respectively". Do the numbers indicate buried surface areas?
8. The authors present biochemical results measuring "NAD⁺ level (A450)". It would be helpful to explain the setup of the assay in Results.
9. Discuss how the structural activation mechanism of SPARSA compares with that of tetrameric SPARTA, which was recently published (<https://www.science.org/doi/10.1126/sciadv.adh9002>).

REVIEWER COMMENTS

Reviewer #1 (Remarks to the Author):

In this manuscript, Zhen *et al* solved the cryo-EM structure of SPARSA complexes at different states and then performed the structure-guided mutagenesis and MD analysis to elucidate the molecular basis for target DNA binding and the recognition and hydrolysis of substrate NAD. Another recent study also reported the cryo-EM structure of SPARSA complex (PMID: 37311833), however, the authors first time revealed the full-length structures of SPARSA, which is a value addition to the understanding of the anti-phage systems. Importantly, the authors further revealed the catalytic mechanism of NADase in the SPARSA complex. Overall, this study is an excellent work representing a major advance in our understanding of NADase activity in anti-phage systems. This work should be of broad interest to the microbiology and biochemistry communities. I recommend its publication in Nature Communications.

Response: We acknowledge the reviewer for recognizing the importance of our study. Indeed, the prokaryotic Argonaute associates with other protein effectors, such as TIR-APAZ (SPARTA) ¹ and Sir2-APAZ (SPARSA) ², which are demonstrated as anti-phage systems to protect the cell from phage invasions. Although the activation mechanisms of MapSPARTA³⁻⁶ and CtSPARTA ⁷ are elucidated, the activation of SPARSA are still not fully understood. In fact, we obtained the cryo-EM structures of the full-length SPARSA and the SPARSA-gRNA-tDNA complex after numerous trials. We believe our study will contribute to understanding the mechanism of action of the SPARSA system.

I have some minor concerns below.

Comments:

1. a brief introduction to the SPARTA system should be included in the abstract section

Response: We followed the reviewer's suggestion and a brief introduction of the SPARTA was added in the abstract of the revised manuscript.

2. The manuscript should be further proofread before publication. Typo and grammar errors should be corrected, e.g. line 46 use should be uses; line 54 eluciate should be elucidate; line 121 assembly should be assemble; line 137 1400 and 1500 should be 1,400 and 1,500

Response: We are sorry for the unintended errors in the previous version of the manuscript. We have proofread the text carefully based on the helpful comments from the reviewers, and all the errors mentioned by the referee are corrected in this version of the revised manuscript.

3. I recommend that the authors move some experiment details to the method section to simplify the manuscript, e.g., lines 120-127 and 262-264.

Response: We followed the reviewer's suggestion. The referred sentences were rephrased in the maintext and some of the contents were moved to the method section accordingly as suggested. Please see line 522-525 and line 542-547.

4. line 130 3.6 Angstrom may not be claimed as high resolution.

Response: We rephrased it as "Finally, we obtained three-dimensional reconstruction of SPARSA at an overall resolution of 3.6 Å".

5. line 140 "Sir2 and APAZ domain, i.e.," should be deleted

Response: Thank you for raising this point. We deleted this, accordingly.

6. Interestingly, another recent study suggests that the Sir2 domain would be flexible after activation, which contrasts with this work. How does the author explain the differences? In my opinion, the flexible Sir2 domain may represent another state of the catalytic cycle.

Response: We thank the reviewer for this helpful information. In the recent publication, Wang *et al* determined the cryo-EM structures of SPARTA-gRNA, SARTA-gRNA-tDNA and SPARSA-gRNA-tDNA. In their study, they mainly focused on the NADase activation of the SPARTA. However, the Sir2 domain in the SPARSA is lacking, which hinders understanding of the activation of the Sir2 NADase.

In fact, the activation mechanism of the Sir2 NADase was only investigated in the ThsA, showing that the activation of its NADase requires the binding of 3'cADPR to the SLOG domain, which leads to the destabilization of the Sir2:Sir2 domain interactions for exposure of the active site for NAD⁺ degradation⁸. The conformational change was observed in the small domain, especially the movement of α 3 helix, enabling NAD⁺ access to the catalytic residues⁸. However, it appears that activation of the Sir2 NADase of SPARSA is different to ThsA on that it can bind NAD⁺ in vivo and in vitro regardless of hydrolysis of NAD⁺⁹, which can also be seen in our NAD⁺ bound SPARSA structure (Fig.3). These findings suggest that activation mechanism of the Sir2 NADase does not involve exposure of the active site for NAD⁺ degradation, which greatly differs from ThsA⁸.

Here, in order to elucidate the potential mechanism of the transitions from inactive NAD⁺ bound state to an NAD⁺ hydrolyzed state upon binding of tDNA and gRNA, we performed molecular dynamics simulations. In the end, a putative mechanism of NAD⁺ hydrolysis was proposed (Fig.6). 7. The NAD density was also observed in the target DNA-bound state. Perhaps consider explaining why the NAD is not hydrolyzed by the activated SPARSA complex.

Response: We thank the reviewer for this helpful comment. At the beginning, we were also confused by this phenomenon. However, we noticed a similar phenomenon observed in a recent paper on the TIR-APAZ system (SPARTA) that NAD⁺ was not hydrolyzed by the SPARTA⁵. We reason that the unhydrolysis might have resulted from the very limited amount of NAD⁺ in our study. We did not add extra NAD⁺, and the identified bound NAD⁺ was endogenous and likely came from protein expression and purification in *E. coli*.

8. The lid region of the SIR2 domain, for example, in ThsA and SIR2-APAZ protein (PMID: 6048923;37311833), is proposed to act as a switch to regulate the NADase activity. The lid region (aa 60-110) in the provided PDB also undergoes a conformational change with high flexibility after the target DNA binding, which is consistent with the previous observations. This point should be mentioned and discussed.

Response: The authors thank the reviewer for this helpful comment. In ThsA, destabilization of the Sir2:Sir2 domain interactions within the tetramer is supposed to trigger its NADase activity caused by the conformational change in the lid region (α 3 in the small domain of Sir2), and to enable NAD⁺ to access the active site. However, our apo form SPARSA can bind NAD⁺, which is distinct to ThsA. We added this section in the discussion now.

9. The NAD density in the target DNA-bound state should be shown. The target DNA-bound structure in this study may represent a post-substrate-bound state, whereas the missing sir2 domain in another recent study may represent a pre-substrate-bound state. It may be beneficial to thoroughly compare and include this information in the manuscript.

Response: Thank you for this helpful and insightful comment. During submission of the manuscript, mechanisms on the activation of SPARTA have been published by seven independent groups. Now, they are cited in this revised version of the manuscript and their activation mechanisms are discussed. The density of NAD⁺ was added in Supplementary Fig.11b as suggested.

Reviewer #2 (Remarks to the Author):

The manuscript by Zhen et al. reports the cryo-EM structure of the SPARSA system, recently demonstrated to be part of defence mechanisms against phage infections in prokaryotes, triggering NAD⁺ depletion upon recognition of invader DNA. The current work elucidates the structure of the SPARSA system in its apo form and in complex with nucleic acids, providing a structural basis for understanding how tDNA binding activates the NADase activity. Key residues are identified by analysing NADase activity of a series of single mutants. Additionally, it employs molecular dynamics simulations to provide insight into the mechanism of NAD⁺ hydrolysis. As such, I believe the authors' work represent a significant contribution to understanding antiphage defence systems in bacteria and the role played by short pAgos. However, before publication several major and minor points should be addressed to increase the confidence in authors' conclusions and clarity of the manuscript.

Response: We thank the reviewer for the positive comment as well as for the valuable suggestions to improve our manuscript.

i. Please add a brief one-sentence explanation of Dali searches. If I understand correctly, they were used for comparative structural analysis against the PDB, but this is not clear from the text. Additionally, a supplementary table summarizing the output from the Dali server should be included (e.g. top 5 nearest neighbours).

Response: Thank you for this helpful comment. Following the reviewer's suggestion, a sentence has been added in line 165-167 to further explain Dali searches. The top five hits of the Dali search have also been added in the supplementary information (Supplementary table 2).

ii. On a related note, the authors identify ThsA as structurally similar to Sir2, however with a relatively high overall RMSD. It would be interesting to compare the Rossmann-like and small domain in isolation to ThsA. Further, the authors note the distances between N142 and H186 and their ThsA analogues differ. I would be curious to know the exact difference since it is hard to judge from Fig. 2e. Perhaps providing a figure with a more local overlap would be more informative.

Response: Thank you for this helpful and insightful comment. According to the reviewer's suggestion, structural superimpositions of the Rossmann-like and small domain have been done separately as shown in Supplementary Fig.4. The difference of N142 and H186 in the SPARSA from those in autoactive ThsA has been further explained in this version of manuscript, and please see line 171-174.

iii. Authors prepared several single mutants that abolished NADase activity. However, whether this is due to the interference with NAD⁺, gRNA or tDNA binding or structural changes within the SPARSA heterodimer is not addressed. If possible, it would be interesting to see analytical SEC chromatograms of SPARSA mutants in comparison to that of the wild type system and binding assays, particularly in the case of N142A and H186A, where MD simulations indicate they are not directly involved in catalysis as suggested earlier in the paper, and double mutant Y89A/R190A.

Response: We thank the reviewer for the concerns. In our study, we introduced mutations on the

residues of SPARSA, responsible either for the NAD⁺ binding, or for the recognition of gRNA and tDNA, but not for the heterodimer formation of SPARSA.

We apologized for the lack of analytical SEC chromatograms in our lab. Fortunately, alternative methods can be used to investigate whether the heterodimer formation of SPARSA is affected by the desired mutations. Since a 6×His-tag was only constructed at the N-terminus of pAgo (there is no His-tag in Sir2-APAZ), the wild type Sir2-APAZ can be co-eluted with pAgo using the Ni affinity chromatography, suggesting the heterodimer formation of SPARSA. The SPARSA mutants with abolished NADase activities were selected to be purified using Ni affinity chromatography, followed by the SDS-PAGE analysis of the corresponding elutions. The results reveal that all mutations have no effect on the heterodimer formation of SPASA, which is in line with our previous conclusion in our original manuscript.

iv. Based on molecular dynamics simulations authors suggest NAD⁺ is cleaved by water molecules that enter the catalytic site after a conformational change in loop1 upon gRNA/tDNA binding. This appears connected to His186-His163 hydrogen bonding. The outcomes of MD simulations are appealing, however performing replica simulations would help to increase the reliability of their findings regarding the loop movement. Additionally, it would be interesting to see the position of the loop1 before MD simulations in SPARSA-NAD⁺ and SPARSA-gRNA/tDNA/NAD⁺ as in panel Fig. 7e, and after MD for N142A and H186A mutants, as well as RMSF of the loop residues for the different systems. Finally, the authors should include a short comment on the mobility of water molecules I and II and how N142A destabilizes the His186-His163 hydrogen bond.

Response: We thank the reviewer for these valuable concerns and suggestions. To increase the reliability of our finding on the movement of loop1, we further performed a replica MD simulation

for different assembled systems including SPARSA and SPARSA-gRNA/tRNA in the presence of Sir2-APAZ WT, H186A, or N142A (3000 ns long in total). The results were highly consistent with those of our original manuscript (Supplementary Fig. 17). We also added information about the position of loop1 and the RMSF of the loop1 residues for different systems as suggested (Supplementary Fig. 16). Moreover, to identify the mobility of water molecules, we optimized bar graphs by employing violin plots, which depicted the distribution of the number of water molecules during MD simulations (Fig. 6b). We additionally added the time evolution of the number of water molecules during MD simulations (Supplementary Fig.15). According to the suggestion, we provided a brief comment on the water mobility and the mechanism of the destabilization of the H186-H163 hydrogen bond by the N142A mutation in our revised manuscript. Please see lines 450-455.

v. Please add details on cluster analysis and how the number of water molecules surrounding NAD⁺ was calculated in the methods section.

Response: We thank the reviewer for the suggestions. The details of cluster analysis and water mobility analysis were added in the methods section of the revised manuscript. Please see lines 608-613 and line 616-618.

vi. Caption to Fig. 7 does not match the figure.

Response: We are sorry for the mistake. Thank you for pointing out the error. Now the caption to Fig 7 (Now Fig.6) has been revised. In addition, all captions are checked.

vii. Fig. 6c: Please add labels indicating the location of the loop as done in Fig. 6b.

Response: We followed the reviewer's suggestion, and the residue labels for the location of the loops have been added.

viii. Finally, the manuscript would benefit from a couple additional rounds of refinement to improve clarity and readability. Certain concepts or structural regions are introduced late in the text or poorly defined (e.g., $\alpha 7$, $\alpha 15$, $\alpha 16$ within the Rossmann domain, $\alpha 11$, $\alpha 12$ within the PIWI domain). I would suggest they be included in the section discussing Fig. 1 along with a figure denoting regions of interest ($\alpha 7$, $\alpha 15$, $\alpha 16$ etc.).

Response: We followed the reviewer's suggestions. The manuscript is polished by an English speaker to improve its clarity as suggested. Since the most important finding in this study is the mechanism of activation of the Sir2 NADase, and the interactions between pAgo and Sir2-APAZ is not the key, we rephrased and simplified discussion of the interactions in the maintext (line 208-230). The original Fig3 are now moved to the Supplementary Figure 8.

Reviewer #3 (Remarks to the Author):

X. Zhen *et al* present structural and biochemical findings aiming to elucidate the mechanism of a *Geobacter sulfurreducens* SPARSA system, consisting of two domains of a short pAgo-family protein, Sir2 NADase and APAZ domains. The theme of SPARSA and SPARTA systems that initiate anti-phage responses via hydrolysis of NAD⁺ is fascinating. The manuscript describes cryo-EM, molecular dynamics and mutational/biochemical data that could shed light on the mechanism of Gs SPARSA activation. Unfortunately, the manuscript in its current form does not achieve the goal of clarifying the mechanism. My regarding the presented data and their mechanistic interpretation are listed below. Furthermore, the manuscript is hard to read due to typographical and stylistic errors. I do not have the capacity to list such numerous errors: there are more than a dozen on each page.

The authors must correct the conceptual and grammatical issues to present their findings in a clear way.

Response: The authors greatly appreciate the valuable and helpful comments from the reviewer that helped improve our manuscript. According to the reviewer's suggestion and the latest publication on SPARTA⁵, we think that the complex of SPARSA-gRNA-tDNA in our study is most likely in the prehydrolyzed state, which is similar to SPARTA⁵. We addressed all the issues pointed out by the reviewer. Meanwhile, we sincerely apologized for the poor language of previous version of our manuscript. In the current version of manuscript, we have worked on enhancing the readability, and have also involved native English speakers for language improvement and corrections.

1. The major concern is whether the presented cryo-EM structures represent a physiologically relevant complex. If the complex of SPARSA with the RNA*DNA duplex activates NAD⁺ cleavage by Sir2, why does the active site of the SPARSA*RNA*DNA complex contain the substrate, rather than the product(s) or a vacant (post-release) site?

Response: Thank you for this helpful and insightful comment. Recently, a few structures of the tetrameric SPARTA, had been published while this manuscript was under the peer review^{3-6,10,11}. We noticed that in one of these studies⁵, the authors determined cryo-EM structure of tetrameric MapSPARTA containing NAD⁺ in the catalytic sites, which was proposed to be a prehydrolyzed state⁵. Therefore, our results may also represent the prehydrolyzed state, which is similar to MapSPARTA⁵. Due to the fact that SPARSA does not require oligomerization for its NADase activity, there are only subtle structural changes observed in SPARSA upon binding of gRNA/tDNA. Unfortunately, we failed to identify other states of SPARSA as reported in SPARTA⁵.

2. The authors describe many interactions in detail, and the comparison of structures with and without the duplex allows them to reveal larger conformational changes. These descriptions are not very helpful if one tries to understand the mechanism of SPARSA activation. It is unclear, how the local interactions are responsible for the large conformational changes upon RNA*DNA binding. Furthermore, are the large conformational changes in Sir2 important for catalysis? (also, what does this non-quantitative description imply: "huge conformational changes of Sir2 domain"?)

Response: Thank you for pointing out this issue. In the SPARTA and SPARSA anti-phage systems, the enzymatic activity of the NADase is controlled by the tDNA binding. The most significance is to reveal the mechanism of SPARTA and SPARSA as the reviewer suggested. Thus, in this version of the manuscript, the interaction between pAgo and Sir2-APAZ is briefly described and simplified, and the original Fig.3 is moved to the Supplementary Figure.

As to the conformational changes in the Sir2 domain, two regions are involved, including the H186 loop and the small domain, of which the former region is demonstrated responsible for the NADase activation in this study. The activation mechanism of SPARSA differs from that of ThsA^{8 12} on that it does not involve exposure of the active site for NAD⁺ that are mediated by the conformational change in the small domain⁸.

3. The abstract contains an exaggerated claim that "functions of prokaryotic counterparts are largely unknown". Researchers studying pAgos might disagree. A more careful statement could be "functions ... are less well understood".

Response: We thank the reviewer for pointing out this issue. We followed the reviewer suggestion and corrected it in this version of manuscript.

4. In the introduction, the authors incorrectly claim that eukaryotic Ago proteins cleave DNA.

Response: We are sorry for the mistake. The statement was corrected in the revised version of the manuscript.

5. What is the “5’-phosphorylate”?

Response: We are sorry for the error. It was corrected in the revised version of the manuscript.

6. What does the statement “expression of GsSir2 and short pAgo alone in *E. coli* led to inclusion” mean?

Response: The operon of SPARSA consists of genes encoding pAgo and the associated Sir2-APAZ (GsSir2). When pAgo and GsSir2 were individually cloned into the expression vector pET28a and expressed in *E. coli*, both overexpressed proteins were only detected in the form of inclusion bodies, which are useless for in vitro protein assays but represent a common phenomenon that is often seen in the expression and purification of proteins. However, we overcame this difficulty and obtained the soluble SPARSA complex by co-expressing GsSir2 and pAgo in *E. coli*.

7. Explain: “The pAgo was clamped tightly by the N-terminal and C-terminal domains of GsSir2 with 1,400 and 1,500 Å², respectively”. Do the numbers indicate buried surface areas?

Response: We apologize for the confusion generated by the previous version of the manuscript. Yes, the numbers here indicated the buried surface areas. We amended the sentence in the revised version of the manuscript. Please see line 136.

8. The authors present biochemical results measuring “NAD⁺ level (A450)”. It would be helpful to explain the setup of the assay in Results.

Response: We thank the reviewer for pointing out this issue. We added the explanation in the results section of the revised version of the manuscript (please see line 157-162).

9. Discuss how the structural activation mechanism of SPARSA compares with that of tetrameric SPARTA, which was recently published (<https://www.science.org/doi/10.1126/sciadv.adh9002>).

Response: We thank the reviewer for this suggestion. We noticed that six research papers on the structural basis for the activation of the tetrameric SPARTA were recently published^{11 3-5} while our manuscript was under peer review. The mechanistic basis for the activation of SPARTA and our results of the SPARSA were now compared and discussed in line 479-483 as suggested.

References

- 1 Koopal, B. *et al.* Short prokaryotic Argonaute systems trigger cell death upon detection of invading DNA. *Cell* **185**, 1471-1486 e1419, doi:10.1016/j.cell.2022.03.012 (2022).
- 2 Vassallo, C. N., Doering, C. R., Littlehale, M. L., Teodoro, G. I. C. & Laub, M. T. A functional selection reveals previously undetected anti-phage defence systems in the *E. coli* pangenome. *Nat Microbiol* **7**, 1568-1579, doi:10.1038/s41564-022-01219-4 (2022).
- 3 Shen, Z. *et al.* Oligomerization-mediated activation of a short prokaryotic Argonaute. *Nature* **621**, 154-161, doi:10.1038/s41586-023-06456-z (2023).
- 4 Wang, X. *et al.* Structural insights into mechanisms of Argonaute protein-associated NADase activation in bacterial immunity. *Cell Research* **33**, 699-711, doi:10.1038/s41422-023-00839-7 (2023).
- 5 Ni, D., Lu, X., Stahlberg, H. & Ekundayo, B. Activation mechanism of a short argonaute-TIR prokaryotic immune system. *Sci Adv* **9**, eadh9002, doi:DOI: 10.1126/sciadv.adh9002 (2023).
- 6 Aggarwal Aneel K, K., Jithesh Malik, Radhika. Nucleic Acid Mediated Activation of a Short Prokaryotic Argonaute Immune System. *Biorxiv*, doi:10.1101/2023.09.17.558117 (2023).

- 7 Lijie Guo, P. H., Zhaoxin Li, Yong-Cheul Shin, Purui Yan, Meiling Lu, Yibei Xiao. Structural basis for auto-inhibition and activation of a short prokaryotic Argonaute associated TIR-APAZ defense system. *bioRxiv*, doi:10.1101/2023.07.12.548734 (2023).
- 8 Manik, M. K. *et al.* Cyclic ADP ribose isomers: Production, chemical structures, and immune signaling. *Science* **377**, eadc8969, doi:10.1126/science.adc8969 (2022).
- 9 Zaremba, M. *et al.* Short prokaryotic Argonautes provide defence against incoming mobile genetic elements through NAD(+) depletion. *Nat Microbiol* **7**, 1857-1869, doi:10.1038/s41564-022-01239-0 (2022).
- 10 Zhang Jun-Tao, W. X.-Y., Cui Ning, Tian, Ruilin, Jia Ning. Structural basis for ssDNA-activated NADase activity of the prokaryotic SPARTA immune system. *bioRxiv*, doi:10.1101/2023.07.14.549122 (2023).
- 11 Guo, M. *et al.* Cryo-EM structure of the ssDNA-activated SPARTA complex. *Cell Research* **33**, 731-734, doi:10.1038/s41422-023-00850-y (2023).
- 12 Ka, D., Oh, H., Park, E., Kim, J. H. & Bae, E. Structural and functional evidence of bacterial antiphage protection by Thoeris defense system via NAD(+) degradation. *Nat Commun* **11**, 2816, doi:10.1038/s41467-020-16703-w (2020).

REVIEWER COMMENTS

Reviewer #1 (Remarks to the Author):

The authors have addressed my previous concerns.

Reviewer #2 (Remarks to the Author):

The authors have addressed my concerns regarding the scientific content of the paper in a satisfactory manner.

Reviewer #3 (Remarks to the Author):

Xiangkai Zhen et al substantially revised the manuscript, but my key criticisms remain unaddressed. The mechanistic implications of this study remain unclear.

1. The authors' biochemical assay shows that SPARSA efficiently cleaves NAD⁺. Yet, both cryo-EM structures (with and without DNA-RNA helix) contain uncleaved NAD⁺. The authors' response to my criticism on this issue was "Recently, a few structures of the tetrameric SPARTA, had been published...., the authors determined cryo-EM structure of tetrameric MapSPARTA containing NAD⁺ in the catalytic sites, which was proposed to be a prehydrolyzed state 5. Therefore, our results may also represent the prehydrolyzed state, which is similar to MapSPARTA 5."

This response is not helpful, as the authors compare their conflicting (biochemical and cryo-EM data) with an unrelated study. The discrepancy among their own data must be resolved.

2. According to Methods, no NAD⁺ was added to cryo-EM samples. Is the NAD⁺ density in cryo-EM maps due to the molecule co-purified with the protein? Is it possible that a different molecule got co-purified?

3. The structures of the apo-complex and the supposedly activated complex with RNA-DNA duplex are very similar. They don't seem to explain the mechanism of enzyme activation by DNA – indeed, the substrate remains uncleaved in both structures. What are the functional implications of these two structures, if they don't report on the activation mechanism?

4. Although the authors extensively edited the paper, it remains full of typos and hard to read. It still requires proofreading.

1. The authors' biochemical assay shows that SPARSA efficiently cleaves NAD⁺. Yet, both cryo-EM structures (with and without DNA-RNA helix) contain uncleaved NAD⁺. The authors' response to my criticism on this issue was "Recently, a few structures of the tetrameric SPARTA, had been published...., the authors determined cryo-EM structure of tetrameric MapSPARTA containing NAD⁺ in the catalytic sites, which was proposed to be a prehydrolyzed state 5. Therefore, our results may also represent the prehydrolyzed state, which is similar to MapSPARTA 5."

This response is not helpful, as the authors compare their conflicting (biochemical and cryo-EM data) with an unrelated study. The discrepancy among their own data must be resolved.

Response: We thank the reviewer for raising this issue. The results of NADase assays in this study revealed that NAD⁺ can be hydrolyzed upon the addition of gRNA and tDNA to SPARSA, which is in agreement with the previous study ¹. However, as to sample preparation for the cryo-EM data collection, the assembly of SPARSA-gRNA-tDNA was mainly performed and preserved at 4 °C before being frozen onto the EM grids. At this experimental condition, the NADase activity could be very low. To support this assumption, the NADase activities of SPARSA-gRNA-tDNA were measured at 4 °C and 37 °C, respectively (Supplementary Fig.11 d, please see Figure 1). The results revealed that there was nearly no reduction in the amount of NAD⁺ when the reaction was done at 4 °C, indicating a low NADase activity of SPARSA. In contrast, nearly no NAD⁺ remained when the reaction was performed at 37 °C. These results suggested that the NADase activity of SPARSA-gRNA-tDNA is greatly affected by the experimental temperature, and was significantly reduced at low reaction temperature. Thus, the cryo-EM structure of the SPARSA-gRNA-tDNA may represent the prehydrolyzed state where the bound NAD⁺ is not cleaved.

Figure1. The NADase activity of SPARSA is affected by the temperature.

2. According to Methods, no NAD⁺ was added to cryo-EM samples. Is the NAD⁺ density in cryo-EM maps due to the molecule co-purified with the protein? Is it possible that a different molecule got co-purified?

Response: We thank the reviewer for the concerns. To confirm whether the molecule captured by our purified SPARSA is NAD⁺, the HPLC-MS analysis was conducted. The purified SPARSA was diluted in the buffer containing 20 mM Tris, pH 8.0 and 150 mM NaCl. The sample was concentrated to 20 mg/mL, then, 30 μL of the solution was incubated at 70 °C for 20 min and

centrifuged for 30 min to remove the unfolded proteins. The supernatants were analyzed by HPLC-MS in parallel with a commercial NAD⁺ standard. The NAD⁺ standard produced a characteristic peak of 663.90 m/z, and the SPARSA also produced a peak at the similar position (663.97 m/z, please see Figure 2). Based on the results of HPLC-MS and biochemical experiments, we conclude that the captured molecule is the endogenous NAD⁺, which has also been observed in a previous study ¹.

Figure 2. HPLC-MS analysis of the purified SPARSA in comparison to the NAD⁺ standard. The characteristic MS peak for the NAD⁺ standard is at 663.97 m/z (left panel), and the major characteristic MS peak for the purified SPARSA is at 663.90 m/z (right panel).

3. The structures of the apo-complex and the supposedly activated complex with RNA-DNA duplex are very similar. They don't seem to explain the mechanism of enzyme activation by DNA – indeed, the substrate remains uncleaved in both structures. What are the functional implications of these two structures, if they don't report on the activation mechanism?

Response: We thank the reviewer for the concerns. Although the apo form of SPARSA can bind NAD⁺, it can be observed from our NADase activity assays that the activation of Sir2 NADase activity depends on the recognition of gRNA and tDNA, which is in line with the previous study ¹.

However, the activation of SPARSA induced by the binding of tDNA and gRNA results from the transition of the Sir2 domain from the NAD⁺-bound state to the NAD⁺ hydrolyzed state, which is different to other NADases, such as ThsA ^{2,3} and the TIR-APAZ/pAgo, whose activation depends on the access to the catalytic sites of NAD⁺ of the homotetramers with NADase-active conformation ⁴⁻¹⁰.

The NAD⁺ in the structure of the SPARSA-gRNA-tDNA is not cleaved, which likely represents the intermediate state. In order to elucidate how the NAD⁺ was hydrolyzed by the recognition of tDNA and gRNA, we performed molecular dynamics simulations. The simulations revealed that the L1 loop (residues H186-D188) nearby the catalytic site moved about 3 Å to NAD⁺ upon gRNA/tDNA binding, which caused the opening of a channel for water molecules to enter the catalytic site for the nucleophilic attack on the C1' of NAD⁺. The result of the molecular dynamics simulations was further supported by our mutagenesis that double-point mutations of Y89A and R190A abolished the NADase activity of GsSir2. Taken together, we put forwards for a putative mechanism on how SPARSA is activated by the target DNA. The potential mechanism of NAD⁺ hydrolysis revealed by the molecular dynamics simulations was shown in Figure 6 in the main text.

4. Although the authors extensively edited the paper, it remains full of typos and hard to read. It still requires proofreading.

Response: We are sorry for the typos. Following the reviewer's suggestion, the language of the manuscript has been proofread by the professional editing service provided by Nature Research Editing Service.

References

- 1 Zaremba, M. *et al.* Short prokaryotic Argonautes provide defence against incoming mobile genetic elements through NAD(+) depletion. *Nat Microbiol* **7**, 1857-1869, doi:10.1038/s41564-022-01239-0 (2022).
- 2 Hogrel, G. *et al.* Cyclic nucleotide-induced helical structure activates a TIR immune effector. *Nature*, doi:10.1038/s41586-022-05070-9 (2022).
- 3 Manik, M. K. *et al.* Cyclic ADP ribose isomers: Production, chemical structures, and immune signaling. *Science* **377**, eadc8969, doi:10.1126/science.adc8969 (2022).
- 4 Aggarwal Aneel K, K., Jithesh Malik, Radhika. Nucleic Acid Mediated Activation of a Short Prokaryotic Argonaute Immune System. *Biorxiv*, doi:10.1101/2023.09.17.558117 (2023).
- 5 Gao, X. *et al.* Nucleic-acid-triggered NADase activation of a short prokaryotic Argonaute. *Nature*, doi:10.1038/s41586-023-06665-6 (2023).
- 6 Zhang, J.-T., Wei, X.-Y., Cui, N., Tian, R. & Jia, N. Target ssDNA activates the NADase activity of prokaryotic SPARTA immune system. *Nature Chemical Biology*, doi:10.1038/s41589-023-01479-z (2023).
- 7 Shen, Z. *et al.* Oligomerization-mediated activation of a short prokaryotic Argonaute. *Nature* **621**, 154-161, doi:10.1038/s41586-023-06456-z (2023).
- 8 Ni, D., Lu, X., Stahlberg, H. & Ekundayo, B. Activation mechanism of a short argonaute-TIR prokaryotic immune system. *Sci Adv* **9**, eadh9002, doi:DOI: 10.1126/sciadv.adh9002 (2023).
- 9 Guo, M. *et al.* Cryo-EM structure of the ssDNA-activated SPARTA complex. *Cell Research* **33**, 731-734, doi:10.1038/s41422-023-00850-y (2023).
- 10 Guo, L. *et al.* Auto-inhibition and activation of a short Argonaute-associated TIR-APAZ defense system. *Nature Chemical Biology*, doi:10.1038/s41589-023-01478-0 (2023).

REVIEWERS' COMMENTS

Reviewer #3 (Remarks to the Author):

The authors did a great job addressing the key remaining questions. The unusually strong dependence of enzyme activity on the temperature, rendering the enzyme inactive at 40C, is curious. I hope that in their future studies, the authors will capture the catalytic rearrangements of SPARSA to explain NAD⁺ cleavage, by performing cryo-EM of the sample prepared at a higher temperature.

The manuscript has substantially improved after the addition of new data and proofreading. I now support the publication of this manuscript.